



# The Gap Between Attitudes and Action Within the Geoscience Community's Response to Natural Hazards

Leila M. Gonzales[1], Christopher M. Keane[1], Richard L. Bernknopf[1]

[1]American Geosciences Institute, Alexandria, Virginia, USA

5    *Correspondence to*: Leila M. Gonzales (lmg@americangeosciences.org)



**Abstract.** With the impacts of climate-related hazards, such as extreme heat, heavy precipitation, drought, flooding, wildfires, tropical storms, and severe weather becoming more intense and frequent, exposure to these hazards continues to increase as population growth expands into areas prone to higher hazard risk such as coasts, wetlands, and wildlands. Despite these trends, adaptation efforts remain a patchwork of local initiatives implemented primarily at the individual and household level and are not enough to keep pace with increasing hazard impacts. Most climate communication strategies have targeted non-expert audiences to raise awareness and increase adaptive behaviours. However, studies exploring how climate scientists are engaging professionally and personally with climate change impacts are rare. A key aspect of this study is that it specifically focuses on geoscientists, a cohort of experts who study and understand the causes, impacts, and risks of natural hazards. Their professional work provides a distinct perspective on the tangible consequences of climate change. This study is part of a larger research project which examined discipline-level engagement (i.e., funding, research, publications) and professional engagement (i.e., teaching, learning, work) across the geosciences. We review these larger trends in discipline-level and professional engagement with natural hazards and extend this line of inquiry with this study to assess the integration of expert hazards knowledge into geoscientists' personal decision-making processes. The results of this study indicated a knowledge-action gap related to hazard engagement that appears to be systemic across the geoscience discipline. This study provides a baseline for future research into evaluation of climate expert behaviours and actions as it relates to climate hazards. It also provides a new communication simulation that can be tested internationally and compared to this study's results. In addition, the simulation can be incorporated into in-person settings to facilitate discussion about climate hazard risk considerations.

**Short Summary.** This study examines discipline-level engagement (i.e., funding, research, publications) with natural hazards across the geosciences, professional engagement (i.e., teaching, learning, work) among geoscientists, and assesses the integration of expert hazards knowledge into geoscientists' personal decision-making processes. The results of this study indicated a knowledge-action gap related to hazard engagement that appears to be systemic across the geoscience discipline.

## 1 Introduction

The impacts of climate-related hazards, such as extreme heat, heavy precipitation, drought, flooding, wildfires, tropical storms, and severe weather are becoming more intense, impactful and frequent (USGCRP, 2023; Seneviratne et al., 2021). Concurrently, exposure and vulnerability of communities to these hazards has increased dramatically as population growth and urban development have expanded into high-hazard areas, such as coastal and wetland areas and wildland areas prone to wildfires (Reimuth et al., 2024; USCGRP, 2023; de Koning and Filatova, 2020; AghaKouchak, 2020; Abebe et al., 2019; de Koning et al., 2019; Lazarus et al., 2018). Consequently, the cost of climate-related hazards continues to increase. Specifically in the US, a single (inflation-adjusted) billion-dollar climate-related hazard occurred every four months on average in the 1980s, and that frequency has increased to once every three weeks on average (USGCRP, 2023). Of particular concern is the increasing frequency and severity of compounding hazard events, extreme events occurring at the same time or in quick





succession in the same geographical region (USGCRP, 2023). Compounding events cause more widespread disruption than

single events and are expected to intensify in the future (AghaKouchak, 2020), thus stressing the adaptive capacities of even the wealthiest of countries (Juhola et al., 2022).

Despite these trends, adaptation efforts and investment in adaptation remain lacking (USGCRP, 2023). Most adaptation efforts are fragmented and implemented at the local level only, primarily among households and individuals (Berrang-Ford et al., 2021; Moser and Pike, 2015). Climate change beliefs and information do not in and of themselves appear to encourage adaptive

behaviours (McNamara et al., 2024; van Valkengoed and Steg, 2019; Hall et al., 2018; Hornsey et al., 2016; Flood et al., 2018; Crookall and Thorngate, 2008), and as Hall et al (2018) reported in their study, pro-environmental behaviours were more prevalent among climate change sceptics than among climate change believers who were more likely to support governmental policies addressing climate change. This knowledge-action gap is further compounded by politicized and polarized environments, a general lack of public understanding of climate change, and a growing apathy driven by feelings of

hopelessness and being overwhelmed (Ouariachi et al., 2017; Moser, 2016; Kiparsky et al., 2012). Individuals tend to conceptualize hazard risk as an uncertain risk to distant locations rather than as a direct, personal threat (Bellamy and Hulme, 2011; Whitmarsh, 2008). As such, direct experience with severe impacts from hazards has been noted to drive adaptation action (Silva et al., 2018; Whitmarsh, 2008), and Otto et al (2020) suggest that experience with more severe and frequent extreme climate events may be a fulcrum for breaking the cycle of inaction and apathy.

Communication strategies to increase climate awareness and adaptive behaviours have evolved from unsuccessful information-deficit models where "experts" conveyed scientific information to "non-expert" audiences (Badullovich et al., 2020; Illingworth and Wake, 2019; Corner et al., 2015) to interactive approaches that capture complexity, challenge existing belief structures, facilitate dialog and reflection, and incorporate local knowledge to increase the relevance of climate change impacts (Galeote et al., 2021; Badullovich et al., 2020; Ouariachi et al., 2020; Flood et al., 2018). Many modern interactive approaches

employ games to enhance engagement in climate change concepts, facilitate group problem solving through simulated scenarios, and reinforce individuals' ability to take meaningful action within their communities (Gargiulo, et al., 2025; Galeote et al., 2021; Ouariachi et al., 2020; Crookall and Thorngate, 2008).

While most communication strategies target non-expert audiences to raise awareness and action more broadly, studies exploring how climate scientists are engaging professionally and personally with climate change impacts are rare. Given that

the geosciences constitute the intellectual discipline that provides the expertise and knowledge to understand the causes, impacts, and risks of natural hazards, the community has the opportunity to lead the way in adaptation and mitigation related to natural hazard impacts. In this paper, we examine the US geoscience community's response to natural hazard events through (a) discipline-level engagement (i.e., funding, research, publications), (b) professional engagement (i.e., teaching, learning, work) among geoscientists, and (c) integration of expert hazards knowledge into geoscientists' personal decision-making

processes. We chose specifically to focus on natural hazard impacts instead of climate change impacts because climate change beliefs are strongly aligned with political ideologies in the US (Nurse and Grant, 2019; Hornsey et al., 2018) and natural





hazards impacts provide tangible and direct experiences from climate change processes whereas climate change itself is generally perceived as a distant issue (Bellamy and Hulme, 2011; Whitmarsh, 2008).

The paper is structured as follows:

- Section 2 provides an overview of the discovery process as we explored the geoscience community's response to natural hazard impacts since 2000. It discusses the trends in research investment and production as well as curriculum resource production as it relates to natural hazards research and pedagogical approaches. This section also highlights the results of direct surveys to members of the geoscience community regarding their experiences with natural hazard
80       events and their respective responses related to habits, research, and teaching.

- Section 3 provides the rationale and methodology for our job-choice risk assessment simulation that was used to determine if geoscientists acted differently from non-geoscientists by using their expert knowledge regarding natural hazards in their decision-making processes when choosing a job in a new location.

- Section 4 discusses the results of the job-choice simulation and what it reveals about the behaviour of both
85       geoscientists and non-geoscientists relative to social risk, natural risk, opportunity, money, and mobility. It also discusses these results in context of the larger research project, explores other potential adaptations of the simulation, and provides suggestions to bridge the gap between general interest and community-wide sustained action on natural hazard impacts.

## 2 Background

The research discussed here is part of a larger study to understand the response of US geoscience academic departments to these natural hazards events as organizations, as educational entities, and as principals in the science of these natural processes. Natural disasters have been documented as being negatively impactful on university education and research activity (Beggan, 2010; Houston, 2017; Wright et al., 2013) and particularly impactful on communities and institutions that serve socially vulnerable populations (Government Accountability Office, 2022). However, geoscience academic departments are in a unique
position within their institution given their expertise and knowledge of the causes, impacts, and risks of natural hazard events. Our goal with this larger study was to document best practices, prospective opportunities, and community-wide initiatives led by geoscience academic departments to leverage hazard events as teachable moments and as opportunities for research, as well as chances to lead the broader communities of their institutions, academia, and society to a more resilient future. Our hope was that this portfolio of knowledge could be used to model potential mitigation of impacts and unique learning opportunities
within higher education, as well as across all formal education levels.

We used a stepwise approach to establish a baseline of how the US geoscience academic community experienced and integrated impacts from natural hazards between 2000 and 2020. First, we identified US geoscience academic departments that had direct experience with natural hazard events during the study time period. Next, we examined federal investment in




hazard research as well as the production of scholarly research and curriculum resources related to natural hazards over the
same period. We used this baseline to inform our direct surveying efforts regarding experiences with hazard events and post-
event recovery.

**2.1 Trends in Hazard-related Research and Pedagogy**

To identify the set of set of geoscience academic departments that experienced direct impacts from natural hazard events, we
integrated data from the US Federal Emergency Management Agency's Integrated Public Alert and Warning System (IPAWS),
the OpenFEMA Disaster Declarations dataset, the National Weather Service's weather warnings archive, and the US
Geological Survey's ShakeMap archive into a single events database. We then mapped the locations of 976 US geoscience
academic departments based on AGI's Directory of Geoscience Departments (Keane, 2021) into the spatiotemporal event data.
The results showed that all US geoscience academic departments were impacted by natural hazard events between 2000 and
2020. Given this result, we calculated the proportion of departments located in areas covered by a disaster declaration to see
if a smaller set of departments with more substantial impacts could be identified. The results of this secondary analysis
indicated there were impacts across most geoscience academic departments from climatic hazards such as severe weather (i.e.,
thunderstorms, tornadoes, winter storms) as well as flooding, hurricanes, and wildfires.

Given the ubiquity of experiences with natural hazard events among US academic departments, we expected that there would
be a response seen in the production of scholarly research and curriculum resources related to natural hazards. We analysed
the American Geosciences Institute's GeoRef bibliographic database to identify peer-reviewed publications published by US-
based first authors between 2000 and 2020 to evaluate core changes in US geoscience academia research portfolios related to
natural hazards research. The GeoRef database contains over 4.7 million references to geoscience journal articles, books, maps,
conference papers, reports, and theses. Our results indicated no signal in changes in research intensity as seen in the published
literature in response to natural hazard events.

We next examined geoscience curriculum resources to identify trends in production of these products relative to impactful
natural hazard events since 2000. We used traditional sources of new curricular resources and ideas within the US geoscience
academic community, specifically titles and abstracts from the Journal of Geoscience Education (JGE) and curriculum
resources from the Science Education Resource Center (SERC) website at Carleton College. We used the phi4 large language
model (LLM) for topical classification to analyse titles and abstracts of articles in JGE to determine if the articles were related
to natural hazards. Using a defined set of natural hazard labels and custom prompting, we leveraged the LLM as an interpretive
tool to classify the titles and abstracts and provide explanations for the classification recommendations. We manually validated
the classifications as a cross-check on the LLM results.  Of the 1,392 articles, 8% from 2000-2019, and 4% between 2020-
2024 were hazard related. Topically, half of these articles were related to multiple hazards, predominantly volcanoes and
earthquakes, followed by floods and severe weather. Drought and earthquakes were the most common hazards in single hazard-
related articles. Next, we cleaned the set of resources from the SERC catalogue to remove duplicate content and non-curricular





related items. The cleaned set of data was comprised of 9,495 curriculum resources, 5% of which were hazard related per our analysis of the content with the LLM. Hazard-related curriculum resources frequently pertained to earthquakes, multiple hazards (i.e., earthquakes, volcanoes, and tsunamis; floods, hurricanes, severe weather, and slides), volcanoes, and floods.

Next, we used the llama3.1:8b-instruct-q8_0 LLM to analyse awards and funding opportunities from the National Science
Foundation (NSF), a major federal funder of US geoscience research. Using a defined set of natural hazard labels and custom prompting, we leveraged the LLM as an interpretive tool to classify the awards and solicitations and provide explanations for the classification recommendations. We manually validated the classifications as a cross-check on the LLM results. Of the 239,260 NSF awards that were active between 2000 and 2019, 3.9% were for hazard-related research and 1% of the 3,023 funding opportunities during this period were hazard related. Hazard-related awards between 2000 and 2019 predominantly
focused on multi-hazard research, earthquakes, volcanoes, and hurricanes. Hazards commonly mentioned together in multi-hazard awards included earthquakes and tsunamis, earthquakes and volcanoes, hurricanes and floods, drought and flood, earthquakes and landslides, earthquakes, tsunamis, and volcanoes, hurricanes and tornadoes, and floods and landslides. The proportion of awards granted for multi-hazard research increased from 10% of all active hazard related awards in 2000 to 28% in 2019. Of those opportunities between 2000 and 2019 that were hazard related, 9 were in response to specific natural hazard
events such as Hurricane Katrina, the Haiti earthquake of 2010, the 2011 earthquakes in Japan and New Zealand, Hurricane Harvey, Hurricane Irma, and the 2018 Hurricane season. Most opportunities focused on multiple hazards, followed by those focused on earthquake and volcanic activity.

Between 2020 and 2024, 1.6% of the 52,355 NSF awards and 5% of the 279 funding opportunities were related to hazards. Hazard-related awards were primarily focused on multi-hazard research, weather hazards, volcano hazards, earthquakes, and
floods. For multi-hazard awards, the hazards most mentioned together were earthquakes, volcanoes, and tsunamis followed by drought and temperature extremes. The 14 opportunities related to hazards between 2020 and 2024 mentioned either multiple specific hazards, or natural hazards more broadly, and were primarily focused on research related to mitigation and adaptation to climate change impacts and related hazards.

## 2.3 Geoscientist Engagement with Natural Hazards

The lack of increased investment and engagement in hazards-related research, scholarly research, and curriculum development during the study period indicated little dynamic for engagement with natural hazards across the geoscience discipline based on events or evolving priorities. As such, we deployed a suite of surveys between 2023 and 2025 to gain an understanding of how, when, and why geoscientists, both within and outside of academia, professionally engage with natural hazards. Our direct surveying resulted in 214 validated responses and our rapid-response surveying resulted in 447 validated responses.

Among US-based geoscientists, engagement with natural hazards occurred primarily during academic degree programs as part of coursework and related student projects. Geoscience faculty frequently reported integrating recent events into their teaching, adapting their lectures, discussions, and assignments to incorporate recent hazard events, from hurricanes and floods to





earthquakes and wildfires. When integrating hazards into their courses, faculty emphasized both the science behind natural hazards and links between the science and the social, economic, and ecological impacts, often tying these concepts into larger conversations around climate change and resilience. Hazard-related research noted by faculty, research staff, and postdoctoral fellows focused primarily on advancing knowledge and theory and work on prevention and planning/outreach. Non-academic geoscientists mentioned engagement through application of their expertise in real-world disaster recovery planning, prevention, and mitigation. The hazards which geoscientists frequently noted professional engagement included floods, earthquakes, and severe weather. Nearly half of faculty surveyed reported that their professional engagement with natural hazards was unfunded, while approximately two-thirds of academic research staff and post-doctoral fellows reported reliance on federal funding for hazard research. Those individuals who provided additional in-depth reflections of their experiences with natural hazards frequently noted the brevity of the impacts, with events usually lasting a day to a few weeks. Additionally, respondents noted that damage to institutional facilities was generally minimal and usually constrained to power outages and flooding, with some mentioning lack of potable water for a couple of weeks. Participants also noted how their experience with hazards inspired research and academic degree trajectories, but these attitudes did not yield hazard-specific actions or outcomes.

Insights from the rapid-response survey highlighted the constructive impacts that experience with hazards had on shaping the academic and career pathways of geoscientists. Survey participants noted how hazards often spurred academic studies, research activities, and choice of geoscience career pathways. Additionally, while many participants mentioned having direct experience with one or more hazards, only a few noted physical impacts from hazards that disrupted their education. Across generational survey cohorts, there was a shift in perspective in terms of the types of hazards mentioned as well as a shift from more local/domestic focus to global. Late-career participants mentioned volcanoes and earthquakes. Mid-career participants mentioned hurricanes, and early career participants mentioned a wider array of hazards around the world as opposed to only US hazards and events. Participants also recognized strong connections between geoscience and societal resilience, particularly in areas such as sustainability, energy, raw materials, human health, infrastructure, finance, and policymaking.

The lack of sustained engagement at the macro-level (i.e., investment, research, scholarly works, pedagogical materials) and the limited, patchy engagement at the micro-level (i.e. professional engagement in teaching, learning, research, work) suggests that the knowledge-action gap is a systemic issue across the geosciences. Beyond engagement during academic coursework, professional engagement with natural hazards is uneven across the discipline. This pattern of fragmented and incremental action also has been documented in other studies showing limited implementation of transformational adaptations to climate change impacts (USGCRP, 2023; Berrang-Ford et al., 2021; Moser and Pike, 2015). We further investigated this apparent knowledge-action gap by developing and deploying an online job-choice risk simulation that was a customized discrete choice experiment (DCE) to test how natural hazard risk, and the perception of that risk influences a person's choice of residence and occupation. Our aim with the simulation was to evaluate if geoscientists prioritized hazard risk in their decision-making more than non-geoscientists and to increase awareness among participants of how they weighed hazard risk in their personal decision-making processes.



## 3 Methodology

### 3.1 Rationale for Discrete Choice Experiment

Discrete choice experiments have been used within the medical community to understand medical practitioner preferences for job characteristics; however, we were unable to identify any use of a DCE that examines job choice preferences and factors natural hazard risk into the job choice features. The study by Zander et al (2020) is perhaps the closest analogue to our research, but their methodology is substantially different. They used a best-worst scaling stated preference method as part of an online survey to understand Australian residents' preferences for specific locations and mobility decisions. They included four

environmental risks (i.e., floods, heat waves, cyclones, and wildfires) and six non-environmental factors (i.e., health care, educational opportunities, crime rate, scenery, living costs, and distance from family and friends) in their study. They found that crime risk was a strong deterrent, while cost of living and health care were strong attractors to a location (Zander et al., 2020). Environmental risks were secondary considerations in the decision-making process, and Zander et al. (2020) proposes that this may reflect a level of self-efficacy to cope with hazard impacts should they occur. In another study, Lu et al. (2015)

conducted a stated preference survey among residents in Bangladesh on changing residence and job locations under flooding and cyclone impacts. Their study included demographic factors (i.e., age, education attainment, size of family, length of residence in current location, size of residential property, annual income), past experience with hazards (i.e., impacts to quality of life, recovery from impacts, relocation due to impacts), consideration of impacts in new job, and impact factors (i.e., frequency, intensity, accessibility). They found that hazard impacts and income, along with other demographic factors, such

as family size and home ownership, significantly affected participants' choice in location change (Lu et al., 2015). This study contributes to the existing literature by using a DCE to assess job choice preferences among geoscientist and non-geoscientist cohorts including natural hazard risk information as part of the job offer information.

### 3.2 Simulation Design

We designed the DCE as an online job-choice risk assessment simulation in which participants were asked to choose between job offers that varied employment factors (e.g., salary, job tasks), risks (i.e., both natural hazards and human factors), and domicile factors (e.g., location of residence, community amenities, etc.). The simulation was comprised of 12 short web pages (Fig. 1 and Appendix A) that guided participants through defining the parameters of their job search, conducting their job choice selection, and then through a set of reflections that allowed participants to think about their consideration of hazard risk

in their personal lives. Reflection topics included questions about participants' experiences with natural hazards, and questions regarding the importance of different factors in job choice and relocation decisions. After completing the reflection questions, participants were asked to complete a set of demographic questions for cohort-based analyses. Participants were then presented



with a set of results regarding their final job choice, hazard sensitivity, and factors important in their decision-making process. The software code and data requirements for the simulation is provided at

https://github.com/AmericanGeosciencesInstitute/GRANDE-simulation. The simulation took an average of 5 minutes to complete.

We designed the first two parts of the simulation (i.e., search parameters, job choice selection) to focus on the job search and selection, with hazard risk information integrated as part of the job offer. By intentionally making the job selection process a primary focus, we were able to assess underlying patterns in actual hazard risk considerations in the decision-making process

during job selection. After the job choice selection was complete, participants were given the opportunity to reflect on their prior experience with hazards, concern for hazards, and their consideration of hazards in their decisions for their current and job locations. The reflection section of the simulation provided participants with a self-assessment of their awareness and engagement with hazard risk. The final screen provided participants with an analysis of their answers to the reflections mapped against their actions relative to perceived hazard risk importance and actual hazard risk at their current and future job locations.

In effect, this section provided a secondary reflective moment for participants to allow them to see if their perception of hazard risk aligned with the choices they made. The results were framed to reveal participants' attitudes and behaviours relative to hazard risk in the hope that the information would challenge participants to consider the implications of their attitudes and behaviours relative to natural hazard risk. Although the simulation was primarily designed as a research instrument and not for pedagogical purposes, the design of the results page follows prior research (Flood et al., 2018; Crookall and Thorngate,

2008) that emphasizes the importance of a debriefing or reflection period that provides participants with the ability to link their actions, behaviours, and knowledge.

The variables in the simulation have been used in other studies: hazard risk and experience (Abebe et al., 2019; van Valkengoed and Steg, 2019; Whitmarsh, 2008; Grothmann and Reusswig, 2006), salary and cost of living (Ronda et al., 2021; Peters, 2017),  crime risk (Zander et al., 2020), concern for hazards (Noll et al., 2022; de Koning et al., 2019; Abebe et al., 2019; van

Valkengoed and Steg, 2019), and protection / adaptation costs (McNamara et al., 2024; Noll et al., 2022). Other studies implementing these variables used household surveys and interviews (Noll et al., 2022; de Koning et al., 2019; Lu et al., 2015; Whitmarsh, 2008), surveys of individuals (Hall et al., 2018; Hornsey et al., 2018), agent-based models (de Koning and Filatova, 2020), and Adaptive Choice-Based Conjoint Analysis models (Ronda et al., 2021; Peters, 2017).





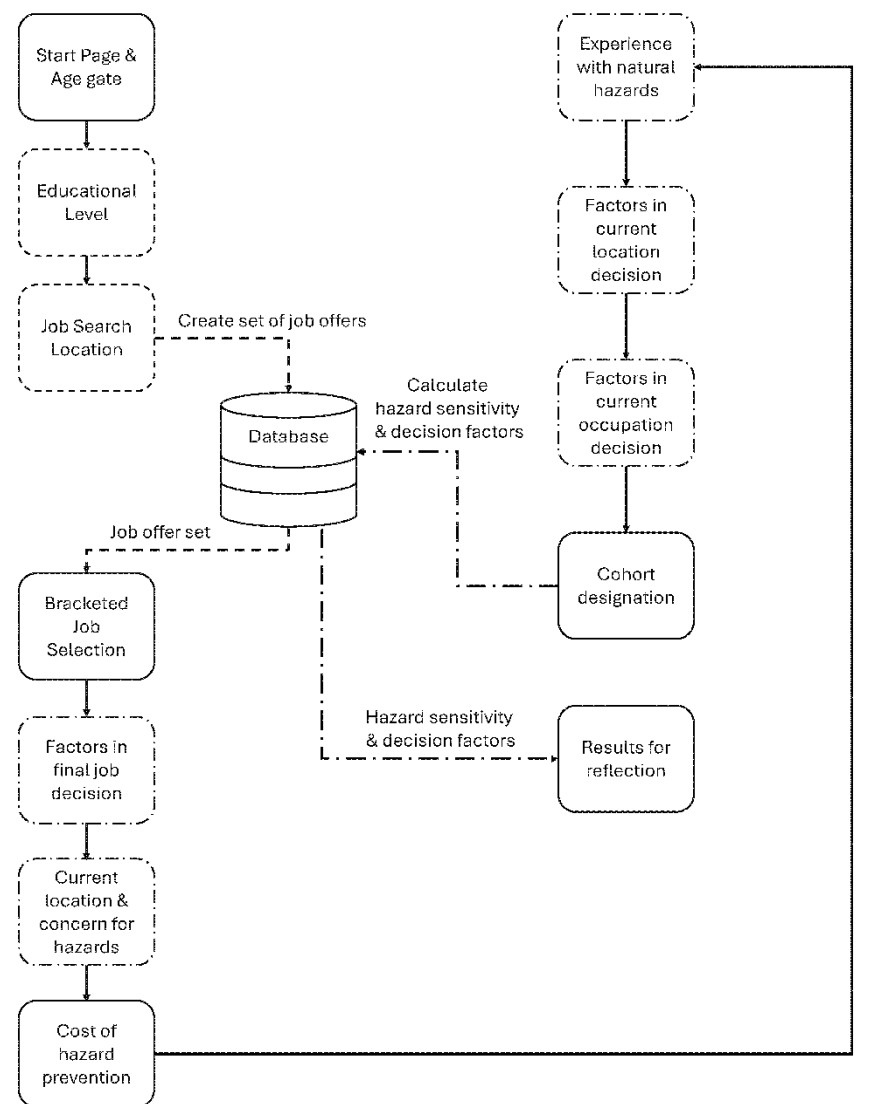


**Figure 1: User interface design.**

**3.2.1 Simulation Data Layer**

To develop the underlying database that was used to generate job offers, we used data from the US Bureau of Labor Statistics'
Occupational Employment Statistics, the US Federal Emergency Management Agency's National Risk Index, the Economic
Policy Institute Family Budget Calculator, the US Federal Bureau of Investigation Crime Data Explorer, NBIRS State Tables,
the U.S. Department of Labor, Employment and Training Administration's O*NET® 28.3 Database, and the US Housing &
Urban Development-US Postal Service ZIP Code Crosswalk dataset to create location-related and job-related tables in our



database. The job tables used information from the US Bureau of Labor Statistics' Occupational Employment Statistics regarding state level salary information for specific jobs. In addition, we used the base occupational titles and descriptions from this dataset to construct a variety of degree field specific job titles and descriptions using the mixtral-8x7b LLM. We first used the LLM to generate a list of topical areas for each degree field and then instructed the LLM to create a job title and description given the BLS title and description as a base with a focus on the topical area. Organization names were created from a randomized assignment of letters from the Greek alphabet and Crayola crayon colour names. The location tables contained the crime risk, hazard risk, cost-of-living data mapped to US cities and counties, as well as to US Census Bureau regions and divisions.

To generate the set of 16 job offers, the simulation used the job search parameters specified by the participant (i.e., highest level of education achieved, degree field of highest level of education, and job search locations). An initial universe of possible locations within the participant-specified search area was constructed. If a participant's search area only included locations in the Western US, we expanded the initial universe of locations to include locations from the West North Central and West South Central US. This was done because locations in the Western US did not meet all the requirements in the job list criteria table (Table 1). The universe of locations included the city, state, county, division, region, crime risk level, crime rate, hazard risk level (both overall and for individual hazards), and cost of living. Next, this universe of locations was filtered into the 8 crime-hazard location zones in the job list criteria table (Table 1), and a subset of locations were randomly selected from these zones to create the initial set of 16 locations for the jobs. Salary to cost of living levels for each location were set based on the job list criteria table. Next, a list of job titles, descriptions, organization names and salary ranges were retrieved based on the job search parameters. The jobs were matched to the locations and the salaries for each job and location was generated randomly based upon the salary to cost of living range as well as the job salary range. This resulted in a set of 16 jobs that met the job list criteria that were within the specified job search parameters.

| Job ID | Salary to cost of living | Crime rate | Hazard risk |
|---|---|---|---|
| 1 | Low | Low - Medium | Very Low - Relatively Moderate |
| 2 | Low | Low - Medium | Relatively Moderate - Very High |
| 3 | Low | Medium - High | Very Low - Relatively Moderate |
| 4 | Low | Medium - High | Relatively Moderate - Very High |
| 5 | Medium | Low - Medium | Very Low - Relatively Moderate |
| 6 | Medium | Low - Medium | Relatively Moderate - Very High |
| 7 | Medium | Medium - High | Very Low - Relatively Moderate |
| 8 | Medium | Medium - High | Relatively Moderate - Very High |
| 9 | High | Low - Medium | Very Low - Relatively Moderate |
| 10 | High | Low - Medium | Relatively Moderate - Very High |





| 11 | High | Medium - High | Very Low - Relatively Moderate |
| 12 | High | Medium - High | Relatively Moderate - Very High |
| 13 | High | Low | Relatively High - Very High |
| 14 | Low | High | Very Low |
| 15 | Medium | High | Relatively High - Very High |
| 16 | High | Low | Relatively Moderate |

**Table 1: Example job list criteria for set of 16 job offers presented to simulation participants.**

### 3.2.2 Simulation User Interface

On the introductory page of the simulation, participants were provided with information about the purpose of the simulation, the steps they would be asked to complete, and the information they would be provided after completing the simulation. They also were provided extra information about the simulation such as the average duration to complete the steps (i.e.,

approximately 5 minutes), the optimal operating systems and mobile devices to use, and age requirements for participation. To begin the simulation, participants were required to specify their age to confirm that they were at least 18 years old.

Next, participants were prompted to define their job search parameters through a series of webpages that allowed them to specify their highest level of education attained, the focus of their highest degree or academic program, and the locations in which they wished to search for jobs. This information was used to generate a list of 16 job offers.

At the start of each round, the game randomized the order of the jobs in the set and then displayed a pair of jobs to the participant, prompting the participant to select either job by clicking on the job they preferred or reject both jobs. For each job offer, the following information was displayed: job title, company name and location, salary, crime and hazard risk, cost of living and disposable income, job description, and job ID (Fig. 2). In Round 1, the game displayed 8 pairs of jobs to the participant. Chosen jobs in Round 1 were randomized and displayed in pairs in the next round (Fig. 3). If there were an odd

number of selected jobs, then after randomization, the first job in the set was also paired with the last job in the set for the round to have a complete set of job pairs. Job selection continued until a final job was chosen. Given this structure, final job choice could occur in Round 1, 2, or 3 depending upon the job selections made by the participant.





**Figure 2: Example of the job offers displayed to the participant in the job selection process (Gonzales et al., 2025).**




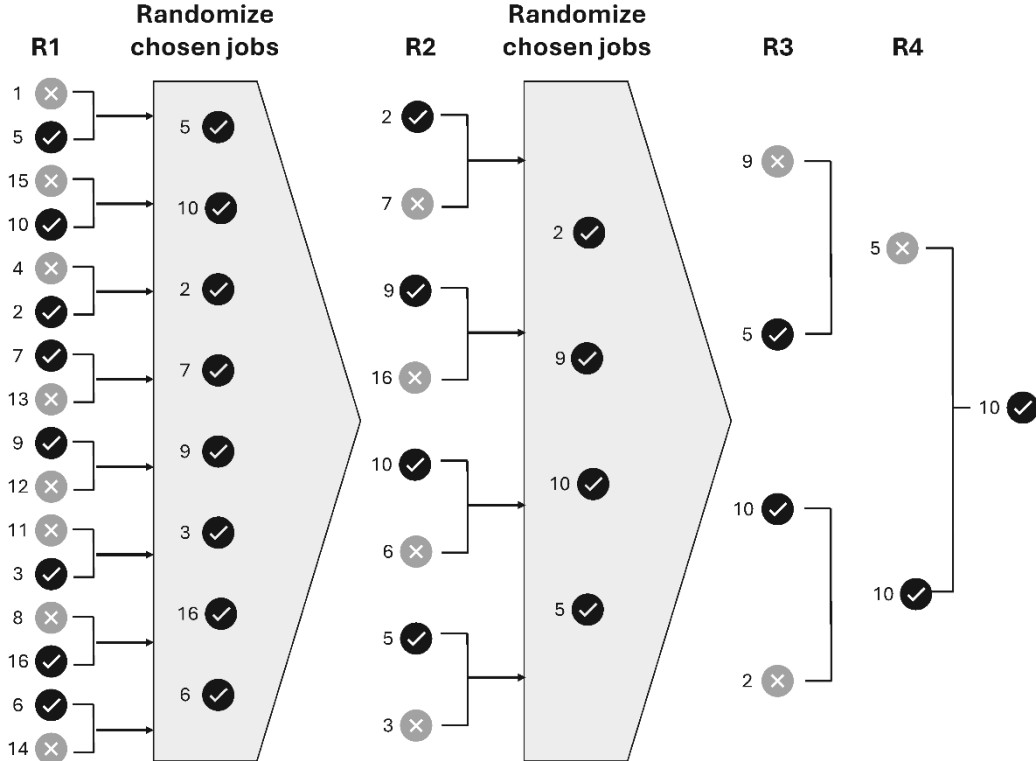

**Figure 3: Bracketed job choice selection process with randomization of selected jobs.**

Once the final job choice was made, the participant was guided through a set of reflection questions pertaining to their decision in choosing the job offer, their experience with hazards relative to their current location, and the factors important to the choice of their current location of residence and current occupation. Lastly, the participant was asked to identify the characteristics of their cohort (i.e., career stage, gender, race and ethnicity). The final page of the simulation provided the user with information about their perception of hazard risk, actual hazard risk, and their ranking of hazard risk against other factors in their decision-

making process.

Participation in the simulation was voluntary and anonymous. The simulation assigned an internal unique token identifier at the beginning of a participant's simulation run. Participant data was recorded initially once a participant completed the job choice selection process and was updated with the reflection and cohort as the participant progressed through the simulation. This part of the process allowed us to capture participant data for those who dropped out of the simulation before making it to

the final results page. Apart from the required questions, such as age, job search locations, and the job search selection process, participants had the option to not answer the other questions in the simulation.





### 3.2.3 Simulation Results for Reflection

The combination of the job choice information and the reflection questions were used to generate the results provided to the participant on the final page, namely information about their hazard sensitivity, defined as the actual hazard risk of the location
and their perception of that risk, as well as information about the factors they said were important in their decision-making processes. The hazard sensitivity information (Fig. 4) provided a combination of text, interactive graphics, and tabular data that discussed the change in sensitivity from the participant's current location of residence to the location of the job they chose. Hazard risk sensitivity was provided as an overall trend and for each hazard type.

Hazard risk perception was calculated for the current location and job location following Eq. (1) and Eq. (2) for each hazard:

$$Ci = \frac{Ch - Is}{Is} \tag{1}$$

$$Rp_d = (0.85 \times Hi_d) + (0.15 \times Ci) \tag{2}$$

where $Ci$ is the concern-impact index, namely the concern for hazards $Ch$ at the current location relative to the experience of hazard impacts $Is$. If $Is$ was 0, then $Ci = Ch$. The concern-impact index was used to determine the amount of under or over-
concern relative to the severity of impacts for a given hazard. For example, a person may have experienced a low impact earthquake event, but was extremely concerned about earthquake hazards, or alternatively a person may experience a severe impact from a severe weather event but was not concerned about severe weather hazards. In the first case, the individual would be overly concerned about earthquake hazards, and in the second example, the individual would be under-concerned about severe weather hazards. $Rp$, the hazard risk perception is the combination of a weighted hazard importance $Hi$ and a weighted
concern-impact index $Ci$ in the current location or location of the job as denoted by $d$. Weights were arbitrarily assigned to 0.85 and 0.15 to up-weight for the importance of hazard risk in the decision-making process and down-weight the concern and prior experience with hazards (i.e., the concern-impact index).

This part of the decision-making process follows the consideration that emotions and past experiences do not generally weigh heavily in the decision-making process (McNamara et al., 2024; Noll et al., 2022; Zander et al., 2020). An average of all
individual hazard risk perceptions was calculated as the average risk perception. Hazard risk perception was calculated for the current location of residence and for the job location and the plotted against the actual hazard risk level for the associated location. Hazard sensitivity zones (i.e. Risk Averse, Risk Aware – Low Risk, Risk Aware – High Risk, Risk Indifferent) were determined based on quadrants from 0 – 2.5 and 2.5 – 5.0 along the risk perception and risk level axes.





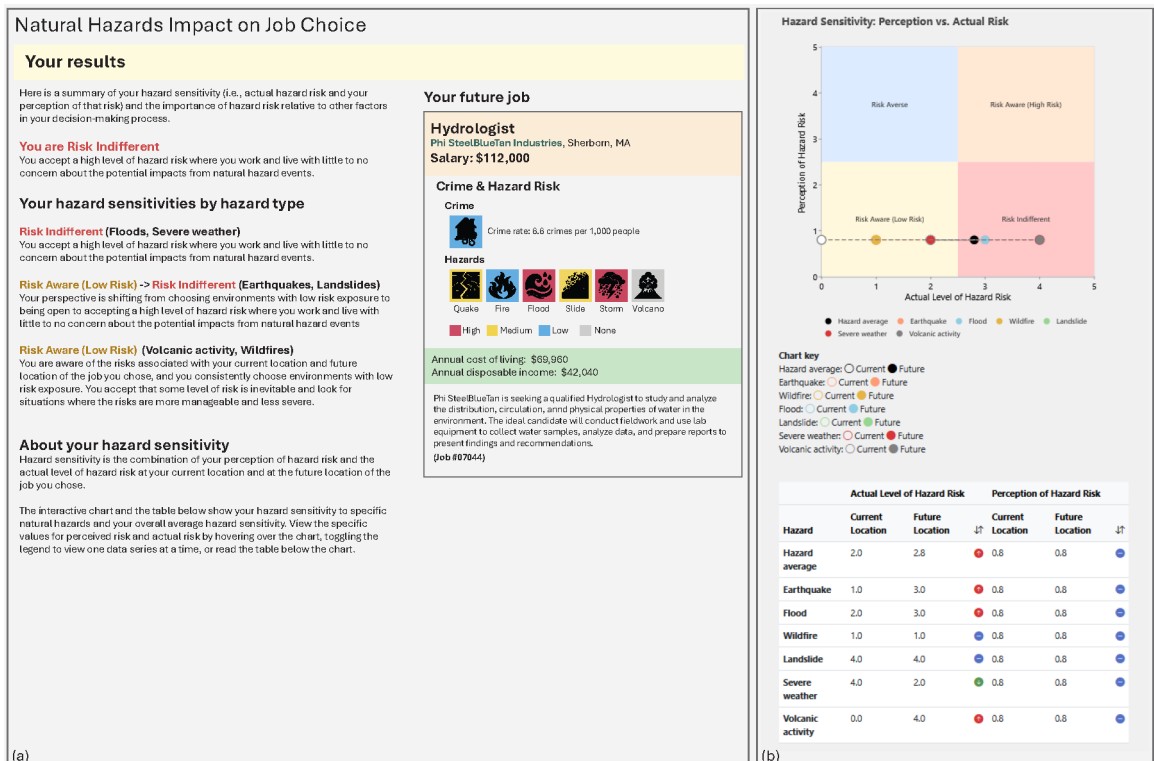

**Figure 4: Simulation results: Hazard sensitivity. (a) Textual description of hazard sensitivity and final job choice information. (b) Graphical and tabular depiction of hazard sensitivity for current location and future job location (Gonzales et al., 2025).**

The participant also was provided information about the factors they said were important in their decision-making processes for their final job choice and their choice of current location and occupation (Fig. 5). Information was provided in textual format as well as in an interactive chart that participants could toggle between the factors influencing their decisions related to their current location and future job location. Hazard risk was displayed as the first factor to highlight it in the list of other factors of importance, such as job characteristics, location, disposable income, crime risk, weather / climate, community amenities, distance from social networks, and other factors.





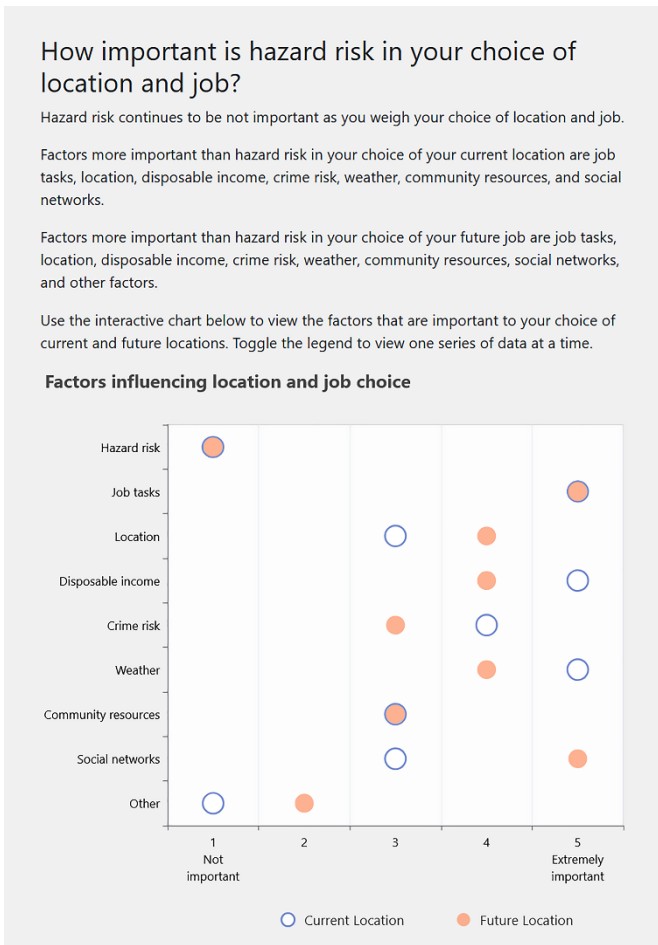


**Figure 5: Simulation results: Factors important to decision making (Gonzales et al., 2025).**

## 4 Results and Discussion

For this research paper, we focused on examining differences between the geoscientist and non-geoscientists cohorts in the simulation. We narrowed our data to those participants that completed the job choice selection process and reflection questions

of the simulation. This resulted in 359 valid responses, 58% from geoscientists and 42% from non-geoscientists. Geoscientists had higher educational attainment than non-geoscientists with 99.5% having a bachelor's degree or higher, compared to 72.8% for non-geoscientists and 88% of the whole population. A Chi-squared test yielded a statistically significant difference between the educational attainment of the geoscientist, non-geoscientist and whole population cohorts ($chi^2 = 66.2325$, p-value $< 0.001$). To understand which of the job features were most valued (i.e., salary-to-cost of living ratio, crime risk, overall hazard risk)

from the set of 16 job offers, we ran a Cox Proportional Hazard model with interaction terms to determine if there were significant differences between the features valued by geoscientist and non-geoscientist cohorts (Table 2). Jobs with higher salary to cost-of-living ratios and lower crime risk were less likely to be eliminated across the job choice rounds. Overall





hazard risk and having a geoscience degree had no effect on job option elimination. Furthermore, the interaction effects (geoscience degree X salary to cost-of-living ratio, geoscience degree X crime risk, geoscience degree X overall hazard risk)

showed that geoscientists valued these job features similarly to non-geoscientists.

| | coeff | exp(coeff) | p |
|---|---|---|---|
| Salary to cost-of-living ratio | -0.42 | 0.66 | <0.005 |
| Crime risk | 0.14 | 1.15 | <0.005 |
| Overall hazard risk | 0.01 | 1.01 | 0.67 |
| Geoscience degree | 0.19 | 1.21 | 0.27 |
| Geoscience degree X Salary to cost-of-living ratio | 0.01 | 1.01 | 0.85 |
| Geoscience degree X Crime risk | -0.02 | 0.98 | 0.51 |
| Geoscience degree X Overall hazard risk | -0.04 | 0.96 | 0.21 |

Table 2: Cox proportional hazard results.

We also ran a logistic regression on the choices made during the final job pair consideration to see if the earlier patterns in job choice selection persisted into the final choice between finalist job offers (Table 3). The salary-to-cost of living ratio remained

a strong and significant predictor of job choice; however, crime risk was not significant in the final choice, likely because high crime jobs had been eliminated earlier in the job choices. Hazard risk was marginally significant and positive, indicating that jobs with higher overall hazard risk were more likely to be chosen from the final pair. We also ran the logistic regression with the same geoscience interaction terms as in the Cox proportional hazard model. This caused all terms to become insignificant except for the salary to cost-of-living ratio and indicate that geoscientists did not behave differently from non-geoscientists on

their final job choice selection.

| | coeff | p |
|---|---|---|
| Salary to cost-of-living ratio | 0.4794 | < 0.001 |
| Crime risk | 0.0225 | 0.802 |
| Overall hazard risk | 0.1394 | 0.066 |

Table 3: Logistic regression on final job choice among final job pairs.

**4.2 Reflections on Job Choice Decisions**

During the reflection section of the simulation, participants were asked to consider the importance of a set of factors in their

decision-making process for their job choice in the simulation as well as for their choice of their location of residence and current occupation. A Chi-squared test showed no statistically significant differences between the responses from the geoscientist, non-geoscientist and whole population cohorts (Table 4); however, there was marginal significance (p=0.0585)





for the importance of job characteristics in the choice of the final job. Given that the geoscience cohort has higher levels of educational achievement than the non-geoscience cohort and that cohorts with higher educational attainment place importance on job characteristics other than salary (Hu and Hirsch, 2017), this marginal significance is to be expected.

|  | Final Job Choice | | Current Location and Occupation | |
|---|---|---|---|---|
|  | chi$^2$ | p-value | chi$^2$ | p-value |
| **Favorable Weather** | 1.6616 | 0.9897 | 6.7797 | 0.5606 |
| **Crime Risk** | 6.5406 | 0.5869 | 9.0222 | 0.3404 |
| **Hazard Risk** | 2.7788 | 0.9475 | 2.946 | 0.9377 |
| **Income** | 1.8492 | 0.9853 | 5.1797 | 0.7382 |
| **Job Characteristics** | 15.0341 | 0.0585 | 12.513 | 0.1297 |
| **Location** | 13.6567 | 0.0912 | 6.0559 | 0.641 |
| **Community Amenities** | 3.8885 | 0.867 | 2.1813 | 0.9749 |
| **Distance from Social Networks** | 6.9451 | 0.5426 | 1.6758 | 0.9894 |
| **Other** | 4.9104 | 0.7671 | 1.7806 | 0.987 |

**Table 4: chi$^2$ test of significance between all participants, geoscientists and non-geoscientists for importance of factors in decision making processes.**

Given the lack of significant difference between cohorts in the importance of factors influencing their decision making, we present the results for all participants in Fig. 6 and Fig. 7. Factors most important in the final job decision included the job characteristics, income, location, and favourable weather. In contrast, top factors of importance in the choice of current location of residence and occupation were job characteristics, location, community amenities, favourable weather, income, and distance from social networks.

Since we conducted an anonymous online simulation, we do not know the exact reasoning for the difference between the factors in these decision-making processes since we could not interview participants directly. However, job characteristics, income, and location ranked as the top three most important factors for participants' current situation and job choice selection. We can propose some hypotheses that may contribute to these differences in how these factors were ranked between participants' current situation and job choice selection.

Part of the difference between the rankings may be in the way we asked the questions. For the question related to factors influencing the choice of current location and occupation, we asked two sets of questions, with the location questions focused on all the same factors as the final job questions, except for job characteristics, and the occupation questions asking only about job characteristics, income, location and other. Another potential reason may be that in the choice of current location and occupation the recall of these decisions may have been tempered by the passage of time. For example, a person may have initially accepted a job primarily due to the increase in pay, but since that time they may have developed a strong network of colleagues and friends and become involved in their community. Thus, in reflection, the importance of income may have





become less important than income. Alternatively, priorities may have changed for the participants, and income may have become more important currently than it was in choosing their current location and occupation.

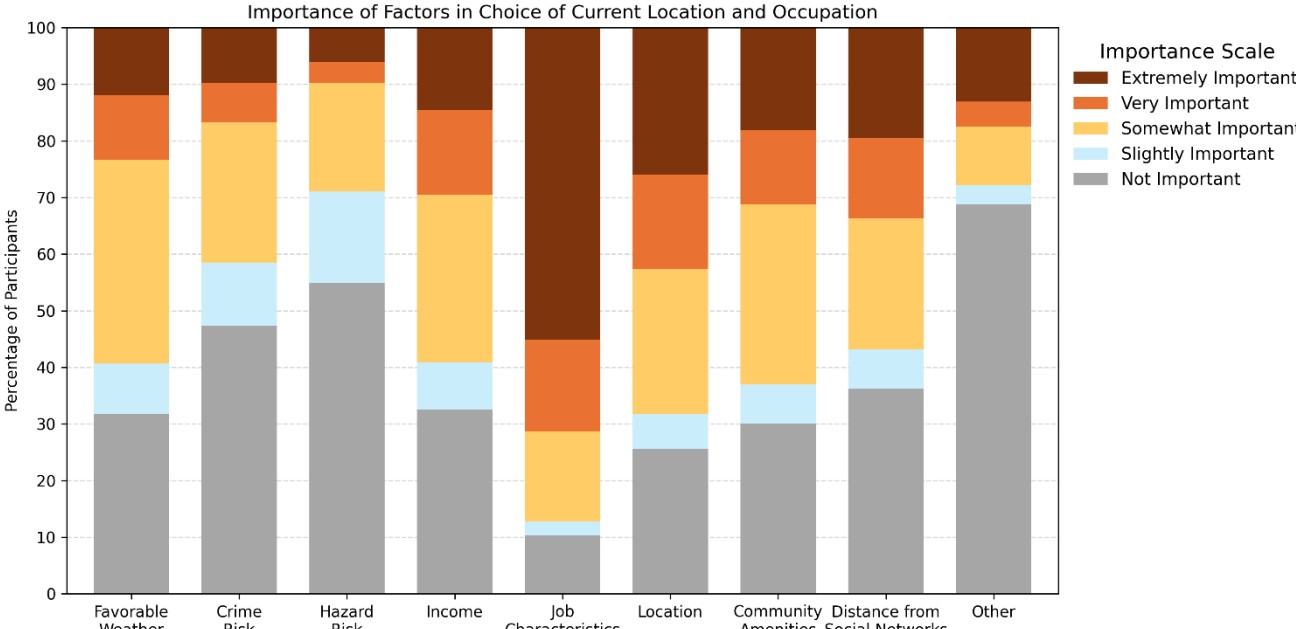

**Figure 6: Importance of factors in job choice decision.**




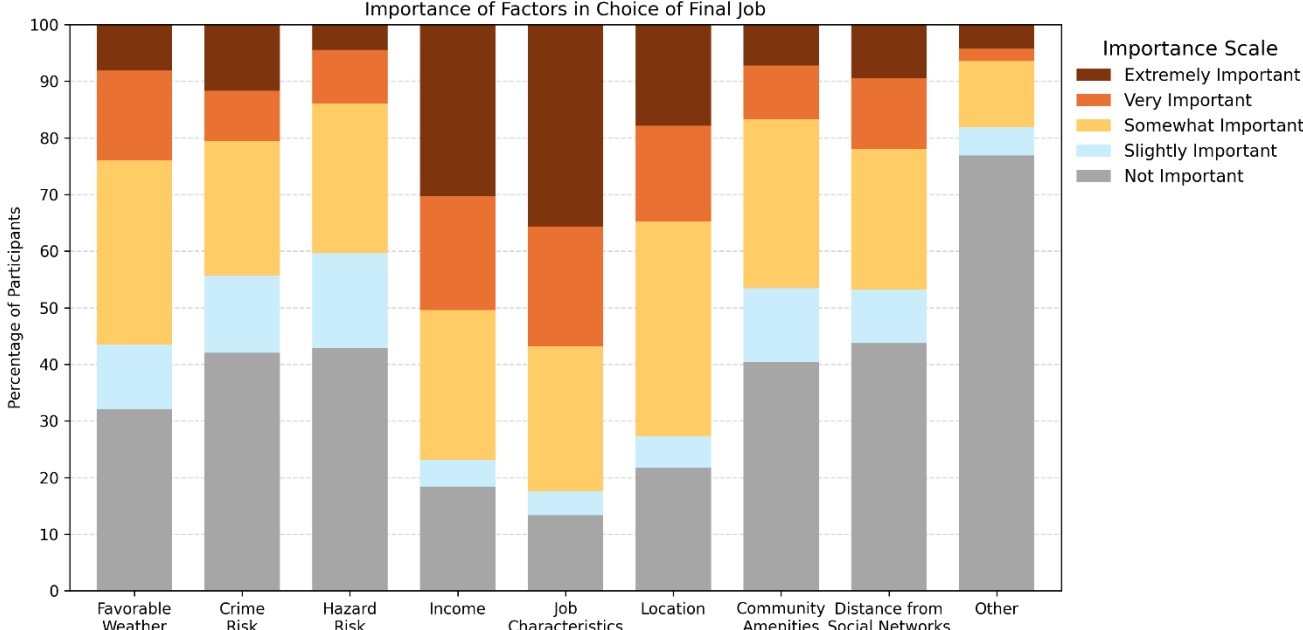

**Figure 7: Importance of factors in choice of current location and occupation.**

We also examined the change in the importance of factors between the final job choice and the current location and occupation, both at the upper levels of importance (very to extremely important) and from the middle to upper levels of importance (somewhat to extremely important) (Table 5). Of note was the 21% increase in participants ranking income very to extremely important in their final job choice in comparison to their current location and occupation. There was also an 11% increase in participants ranking hazard risk as somewhat to extremely important in their final job choice in comparison to their current

location and occupation. However, it is important to note that hazard risk ranked next to last in importance for the final job choice and current location decisions.

| | Very Important to Extremely Important | Somewhat Important to Extremely Important |
|---|---|---|
| **Favorable Weather** | 1% | -3% |
| **Crime Risk** | 4% | 3% |
| **Hazard Risk** | 4% | 11% |
| **Income** | 21% | 18% |
| **Job Characteristics** | -14% | -5% |
| **Location** | -8% | 4% |
| **Community Amenities** | -14% | -16% |
| **Distance from Social Networks** | -12% | -10% |





| Other | -11% | -10% |
|---|---|---|

**Table 5: Change in importance of factors in between choice of final job and choice of current location and occupation.**

Factors important to participants in their decision-making processes related to job choice were consistent with respect to

income and hazard risk, but not crime risk. Income was a strong driver for job choice and was ranked as somewhat to extremely important by 77% of participants. Crime risk was a strong deterrent in job choice but was ranked as somewhat to extremely important by 44% of participants. Hazard risk was marginally significant and positive at the final stage of job choice selection, suggesting that participants were willing to move into higher hazard risk areas given a high enough salary and low enough threshold for crime risk. Hazard risk was rated as not important by 43% of participants and only slightly important by 17% of

participants.

### 4.3 Reflections on Hazard Experience and Concern

When reflecting on the severity of impacts experienced from different hazards, most participants reported either no impacts or low severity impacts from hazards (Table 6). Among hazards, participants reported experiencing higher severity impacts from severe weather and floods, with 38% reporting medium to high severity impacts from severe weather and 19% reporting the

same level of impacts from floods. Furthermore, severe weather and floods were also the two hazards where participants expressed the most concern. A Chi-squared test yielded no statistically significant differences between the responses from the geoscientist, non-geoscientist and whole population cohorts (Table 7).

|  | Severity of Impact Experienced | | | | Concern at Current Location | | | | |
|---|---|---|---|---|---|---|---|---|---|
|  | None | Low | Medium | High | None | Slight | Some | Very | Extreme |
| **Earthquakes** | 64% | 26% | 7% | 3% | 64% | 11% | 13% | 6% | 6% |
| **Wildfires** | 62% | 23% | 9% | 5% | 54% | 11% | 18% | 8% | 9% |
| **Floods** | 47% | 33% | 15% | 4% | 38% | 19% | 31% | 7% | 6% |
| **Severe Weather** | 30% | 33% | 27% | 11% | 24% | 14% | 35% | 13% | 15% |
| **Slides/Debris Flows** | 81% | 13% | 5% | 1% | 69% | 11% | 14% | 3% | 3% |
| **Volcanoes** | 96% | 2% | 1% | 1% | 90% | 5% | 2% | 1% | 3% |

**Table 6: Concern for hazards at current location of residence and severity of impacts experienced from hazards.**


|  | Concern for hazard at current location | | Severity of impacts experienced from hazard | |
|---|---|---|---|---|
|  | chi$^2$ | p-value | chi$^2$ | p-value |
| **Earthquakes** | 4.0541 | 0.8522 | 2.3514 | 0.8847 |
| **Floods** | 5.6093 | 0.6909 | 10.1093 | 0.1201 |
| **Severe weather** | 3.6675 | 0.8858 | 4.3859 | 0.6246 |





| | | | | |
|---|---|---|---|---|
| **Slides / Debris flows** | 1.4416 | 0.9936 | 6.8788 | 0.3322 |
| **Wildfires** | 0.7631 | 0.9993 | 1.6355 | 0.9500 |
| **Volcanoes** | 4.4717 | 0.8123 | 2.7793 | 0.8360 |

**Table 7: chi² test of significance between all participants, geoscientists and non-geoscientists for concern for hazards and experience with hazard impacts.**

We calculated the concern-impact index for each participant as a precursor to our calculation of their hazard perception and assignment of hazard sensitivity zones (Table 8). Most participants had neutral concern-impact values for volcanoes, slides, and earthquakes, and half of participants had neutral concern-impact values for wildfires. Neutral concern-impact index values indicated that concern for the hazard was aligned with the severity of impacts from that hazard. Climate-hazards such as severe weather, floods, and wildfires however had the highest percentages of participants with positive concern-impact index values, indicating that their concern for these hazards outweighed their actual experience with impacts from these hazards.

| | **Negative** | **Neutral** | **Positive** |
|---|---|---|---|
| **Earthquakes** | 15% | 58% | 27% |
| **Wildfires** | 10% | 50% | 40% |
| **Floods** | 12% | 43% | 45% |
| **Severe weather** | 11% | 38% | 51% |
| **Slides / Debris flows** | 8% | 65% | 27% |
| **Volcanoes** | 1% | 89% | 10% |

**Table 8: Concern-impact index. Negative values indicate that impact severity from a hazard was higher than concern for the hazard at the participant's current location of residence. Positive values indicate that concern for the hazard was higher than the level of impacts experienced. Neutral values indicate that the concern was in alignment with the level of impacts experienced.**

## 4.4 Trends in Hazard Sensitivity

Hazard risk perception was calculated as a combination of the concern-impact index and the actual hazard risk level for the hazard at a given location (Table 9). For each location (i.e., current location of residence and final job location), the hazard risk perception was plotted against the actual risk for each specific hazard. Hazard risk sensitivity zones (i.e. Risk Averse, Risk Aware – Low Risk, Risk Aware – High Risk, Risk Indifferent) were determined based on quadrants from 0 – 2.5 and 2.5 – 5.0 along the risk perception and risk level axes (Fig. 8). Risk perception was strongly driven by the level of importance assigned to hazard risk in the decision-making process given the 0.85 weight assigned to this factor in the in Eq. (2). Since most participants rated hazard risk as not important or slightly important, hazard risk perception was generally low among participants for both their current location and at the location of their future job. This outcome resulted in most hazard sensitivities varying between the Risk Aware – Low Risk and Risk Indifferent zones depending upon the actual hazard risk level of the location. Interestingly, perception of hazard risk was higher for the final job choice than for the current location of




residence. This suggests that while hazard risk may not have previously been part of the decision-making process, some

participants did place a higher importance on it when considering their future jobs.

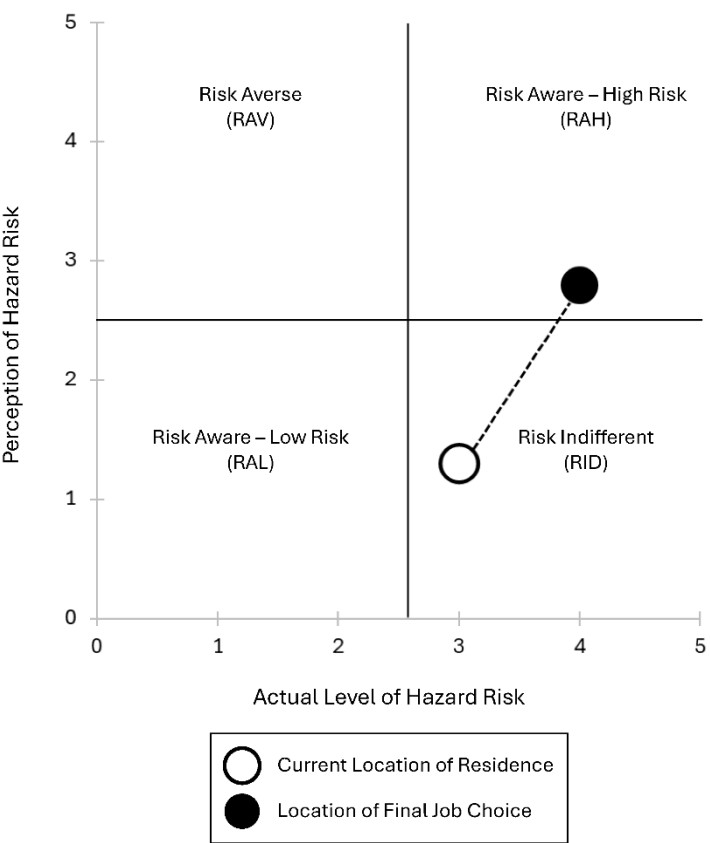

**Figure 8: Example of hazard sensitivity plot for current and future job locations.**

| | Perception of Hazard Risk at Current Location | | | Perception of Hazard Risk at Future Job Location | | |
|---|---|---|---|---|---|---|
| | < 2.5 | 2.5 | > 2.5 | < 2.5 | 2.5 | > 2.5 |
| Earthquakes | 75% | 0% | 25% | 64% | 1% | 35% |
| Wildfires | 73% | 0% | 27% | 62% | 0% | 38% |
| Floods | 73% | 0% | 27% | 63% | 0% | 37% |
| Severe weather | 73% | 1% | 26% | 63% | 1% | 36% |
| Slides / Debris flows | 72% | 0% | 28% | 62% | 0% | 38% |
| Volcanoes | 71% | 0% | 29% | 60% | 0% | 40% |



**Table 9: Perception of hazard risk by participants for each hazard. Lower values (< 2.5) indicate a lower importance placed on hazard risk in decision making. A higher value (> 2.5) indicates a higher importance placed on hazard risk in the decision-making process.**

Next, we plotted the hazard perception against the actual hazard risk for each location to determine the hazard sensitivity zone

of participants. Figure 9 depicts this information for all participants by each specific hazard. For earthquakes and volcanoes, participants showed a decrease in risk aware – low risk (RAL) sensitivity and an increase in risk averse (RAV) sensitivity, indicating that while the hazard risk did not appreciably increase between the current location and future job location, the concern and importance of risk in the decision-making process did. For wildfires and slide hazards, there was a notable increase in risk aversion (RAV) and decline in risk indifference (RID), indicating a shift for some participants from higher to lower

hazard risk with a corresponding increase in concern and importance of risk in the final job choice. Floods, severe weather, and slide / debris flow hazards maintained the highest percentage of participants who remained risk indifferent (RID) for both their current and final job locations; albeit the percentage of participants in the risk indifferent (RID) zone declined between the current and final job locations in all cases, with severe weather and slides showing the largest declines. For floods, those in the risk aware – low risk (RAL) zone and the risk indifferent (RID) zone declined, while those in the risk aware – high risk

(RAH) zone increased. For severe weather, the decline in those in the risk indifferent (RID) zone was matched with an increase in those in the risk aware – high risk (RAH) zone. This suggests that flood and severe weather hazards may be more acceptable to participants than other hazards, even though these hazards had the highest percentages of participants reporting the most concern and impacts compared to other hazards.





Earthquakes     Wildfires     Floods

Severe Weather     Slides / Debris Flows     Volcanoes

Current Location     Future Job

Hazard Sensitivity Key

- ☐ RAL   Risk Aware – Low Risk
- ■ RID   Risk Indifferent
- ■ RAV   Risk Averse
- ■ RAH   Risk Aware – High Risk

**Figure 9: Hazard sensitivity transitions of participants.**

It is notable that only 11% of participants noted high severity impacts from severe weather. This outcome along with other data from the larger research project surveys indicate that within the geosciences specifically, experience with hazards has been relatively short in duration (a few days to a couple of weeks) and low in terms of impact severity (i.e., power outages, minor flooding, etc.). These experiences with natural hazard events may give rise in part to the indifference to hazards of participants in the job simulation. Other studies (McNamara et al., 2024; Noll et al., 2022; Bellamy and Hulme, 2011; Whitmarsh, 2008) have noted that unless individuals experience high severity impacts from hazards, prior experience and concerns have no influence on behaviour. Additionally, given that salary was a significant driver in job choice and an important factor in the job choice decision process, there also may be an assumption that one can finance one's way out of harm, at least to a certain extent. This assumption is supported by other studies that indicate individuals perceive hazard risk as manageable and thus a low-level risk (McNamara et al., 2024; Weber et al., 2022; de Koning and Filatova, 2020, Kiparsky et al., 2012). Specifically, floods and severe weather may be seen as more acceptable hazards in the US, in part due to the ability to obtain hazard insurance and the availability of federal government assistance that aids individuals impacted by natural hazard events in their recovery process. In addition, updated building codes have allowed society to engineer increased resilience to hazards,





allowing individuals to live in areas where there are relatively high risk of severe weather and earthquakes. However, more recently, the ability to obtain hazard insurance in the US has become more difficult (Marcoux and Wagner, 2023) as insurers leave US states where hazard impacts are high, and discussion in the US federal government has indicated intent to reduce or eliminate the Federal Emergency Management Agency (FEMA). In the US, FEMA is a key provider of government assistance to impacted communities. This change in policy raises the question of whether a weakened social safety net will cause a shift

in hazard risk perception and action if federal hazard insurance and recovery assistance efforts become unavailable.

The consistent lack of increasing engagement with natural hazards across the discipline at the macro level, fragmented engagement at the micro level, and lack of integration of expert knowledge into personal decision making indicates that across the geosciences, natural hazards are functionally an "academic" concept rather than a direct, personal threat (Bellamy and Hulme, 2011; Whitmarsh, 2008), despite the increase in frequency and severity of impacts from these events (USGCRP, 2023;

Seneviratne et al., 2021). Beyond engagement with natural hazards during academic coursework, professional engagement with hazards among geoscientists is inconsistent.

The simulation provided participants with a mirror to understand how risk influenced their decision-making processes and revealed that hazard risk is generally ranked low in importance in decision making related to relocation regardless of concern for and prior experience with hazard events. Interestingly, although hazard risk was generally ranked low in importance overall,

there was an increase in hazard perception values across all hazards for the location of the future job in the simulation, indicating that whereas hazard risk may not have entered into the decision-making process previously, some participants did place a higher importance on it when making their final job choice.

### 4.5 Limitations and Future Directions

Given the nature of the anonymous online simulation, it is not possible to know how far the participants considered the

simulation to be a valid representation of reality and how far their behaviour in the simulation could be considered as a faithful representation of what their behaviour would have been in a real-life situation. Using the simulation as part of a larger in-person group discussion would reveal how participants perceived their performance in the simulation with what they would do in a real-world situation. A possible use of the simulation would be in a classroom or with peers whereby individuals would discuss their results after completing the simulation. This type of discussion would provide a way to reflect and debrief with

peers about the decisions made, the importance of hazards in the job selection process, the implications of attitudes and actions, and how individuals perceive hazard risk. Coupling the simulation, in-person discussion and debriefing session, and a longitudinal survey to assess the persistence of the knowledge-action gap among participants could provide a way to measure the impact of this intervention in raising awareness of the knowledge-action gap and promoting adaptive behaviours.

Another limitation to the study is that it was focused solely on the US and may reflect specific socio-cultural patterns related

to job choice patterns and engagement with climate-related hazard impacts that are specific to the US. Running the simulation with populations from other countries may reveal different socio-cultural norms than seen in the US in terms of the importance of hazard risk in job choice selection. Would participants in other countries show the same underlying trends in their job





selection with salary as a significant driver in job choice, crime as a significant deterrent, and hazard risk only marginally significant at the final job selection?

Additionally, running the simulation once new policies related to FEMA are in-place and the market adjusts to the reduction of federal government insurance and mitigation of hazard impacts, will individuals' decision-making begin to change in the US? Since we see increased risk tolerance with higher incomes, potentially proxying a view of sufficient self-insurance when coupled with then-present government support, will there be a change to either substantially higher income thresholds for hazard risk acceptance? Or might there be an elevated the risk awareness across the population?

## 5 Conclusions

This paper reveals that while vicarious and low-impact experiences with natural hazards can spark inspiration to pursue academic studies, careers in the geosciences, and new directions for academic research, sustained action among the geoscience community in response to natural hazards is lacking. Given that financial incentives are key for incentivizing action, we suggest that long-term sustained investment to support hazard related research would provide the necessary response among the geoscience community, spurring research, scholarly publications, and engagement with hazards beyond academic coursework. Furthermore, from an employer's perspective, the job choice risk assessment simulation indicates that employers in high hazard risk areas could attract talent by financially offsetting the hazard risk by offering high salaries. Alternatively, incentivizing workers to move to areas with lower hazard risk exposure would also need to provide salary incentives that would be high enough to offset the non-risk factors that are valued by potential employees such as favourable weather and meaningful work.

While this study reveals that the knowledge-action gap persists within the geoscience community, there are actionable steps that could be implemented to unleash the intellectual capacity of the geosciences to lead the way in adaptation and mitigation, enhancing educational and research opportunities, and building cross-disciplinary networks across disciplines and industry sectors to advance resilience to hazards within communities and across society.



**Appendix A: Simulation Screens**

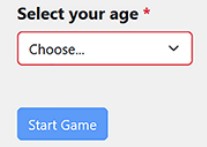

## Natural Hazards Impact on Job Choice

### About the Game

The purpose of this game is to understand how natural hazard risk and the perception of that risk influences a person's choice of residence and occupation. Hazard risk is defined as the probability of a hazard occuring and the value of the property at risk from that hazard.

In this game, you will be asked to choose between job offers that vary employment factors (e.g., salary, job tasks), risks (i.e., both natural hazards and human factors), and domicile factors (e.g., location of residence, community amenities, etc.).

### The Approach

**Step 1:** You will be asked to specify your educational background and location preference.

**Step 2:** Based on your choices in Step 1, a collection of 16 jobs will be selected for you to consider. You will be shown 2 jobs at a time and will be given the option to choose one of those jobs or neither of those jobs.

- If you reject all jobs in the initial collection, 16 new jobs will be generated for you to consider.
- If you select more than one job, you will progress to additional rounds where you will compare the selected jobs until only one job remains.

**Step 3:** After you reach a final job selection, you will be asked about what factors influenced that decision.

**Step 4:** You will be asked about your current location and occupation relative to hazards. This information will be used to calculate your level of hazard sensitivity. Specifically, you will be asked about:

- your level of concern with hazards in your current location,
- the amount you would spend to avoid harm or loss from hazards,
- your prior experience with hazards, and
- the level of importance you assigned to hazard risk in choosing your location and new job

**Step 5:** You will be asked about your career stage and racial, ethnic and gender identities.

**Step 6:** You will be shown your final job choice and information about your hazard sensitivity and factors that influenced your choice of job and location.

This game is part of AGI's Geoscience Program Adaptation to Natural Disruptive Events (GRANDE) research project which is examining how the geoscience discipline has leveraged natural hazard events for enhanced educational and research opportunities.

Read more about our approach on our methodology page.

### Play the Game

Thank you for participating in this research project.

- **To restart the game at any point, refresh your browser.** You will return to this page to start the game again.
- **On average, this game takes approximately 5 minutes to play.** It may take shorter or longer based upon how quickly you arrive at your final job choice.
- **You must be at least 18 years old to play this game.**
- **For the optimal experience, we recommend using a desktop computer. If you're using a mobile device, iPhone 14 or newer, or Android phones provide the best performance.**

**Select your age** *

Choose... ⌄

Start Game


**Figure A1: Introductory screen of online simulation (Gonzales et al., 2025).**




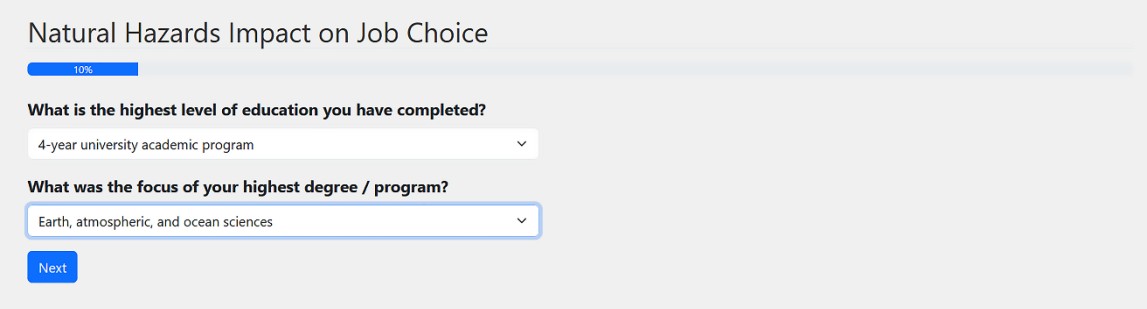

**Figure A2: Educational attainment parameters (Gonzales et al., 2025).**


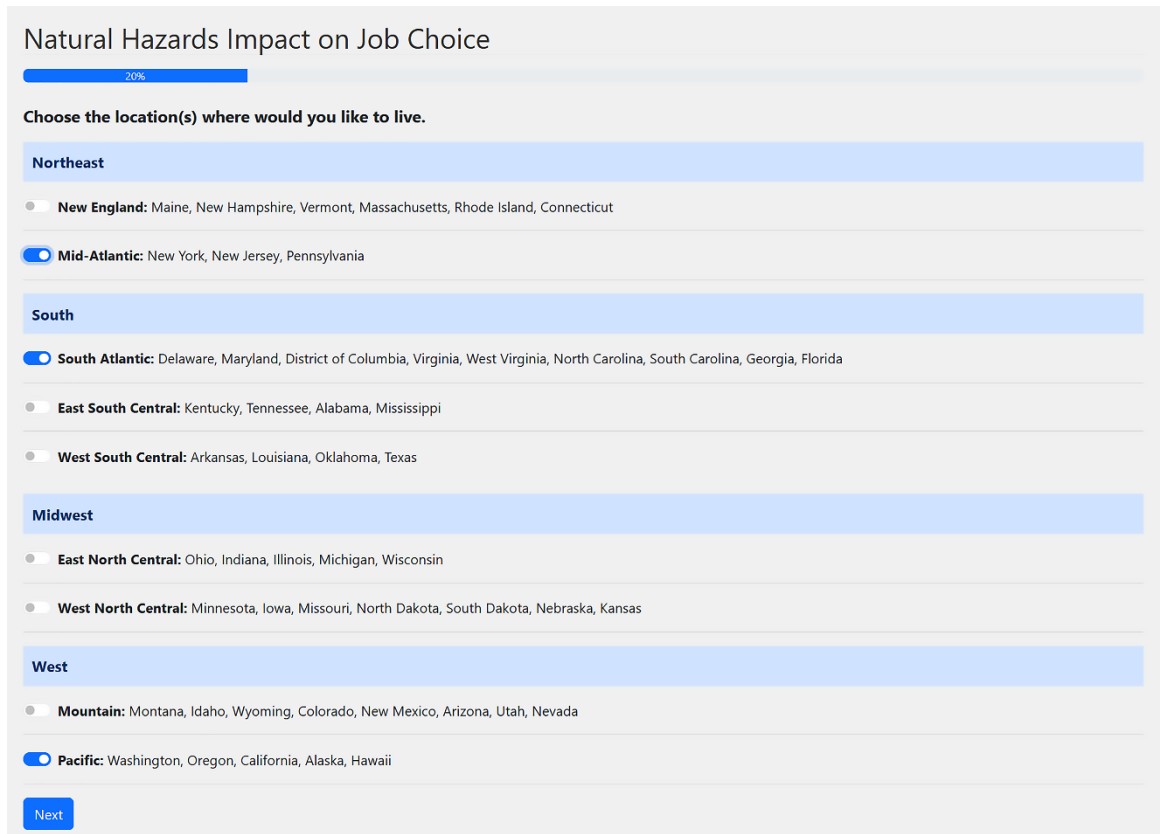

**Figure A3: Job search parameters (Gonzales et al., 2025).**





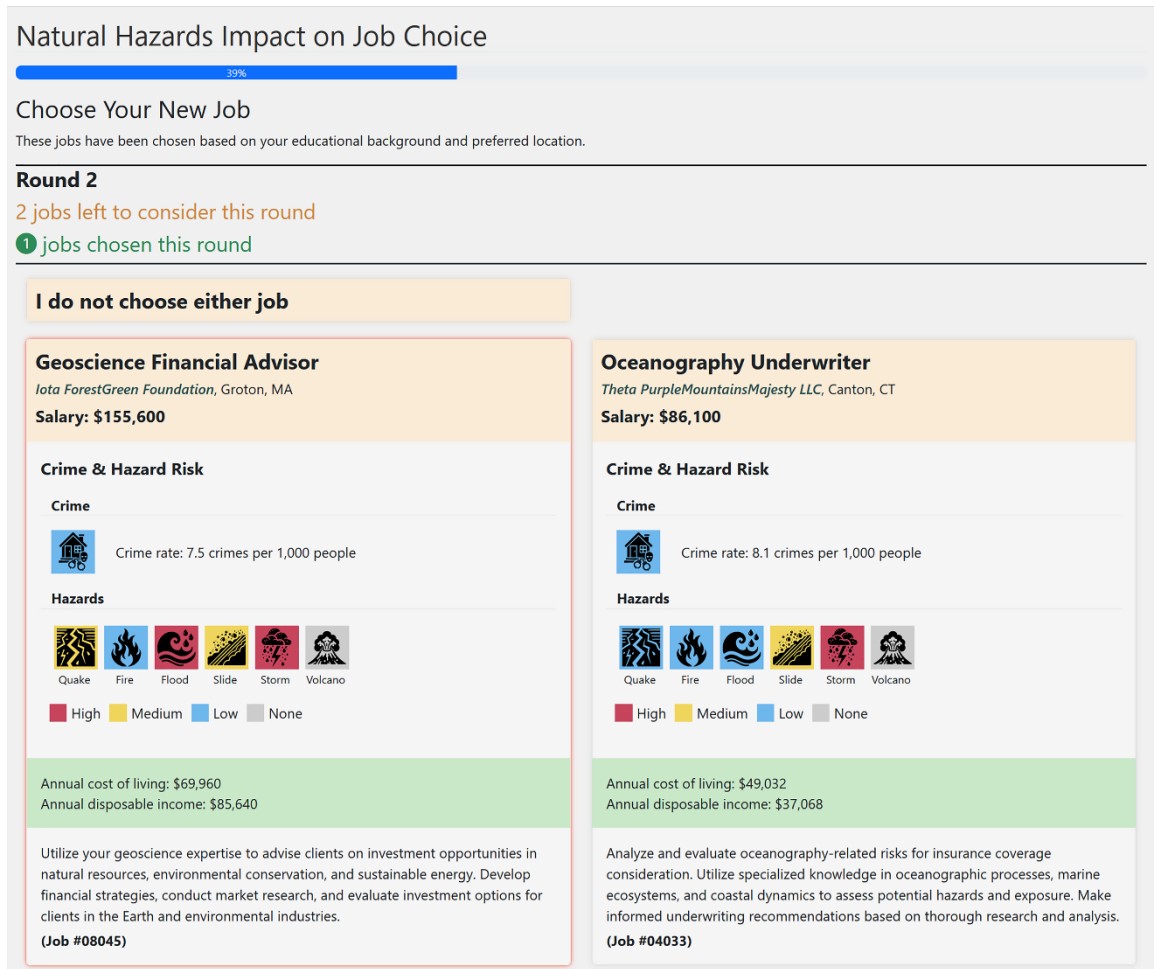

**Figure A4: Example of the bracketed job choice selection process (Gonzales et al., 2025).**



**Figure A5: Reflection on the factors important in the job choice decision (Gonzales et al., 2025).**





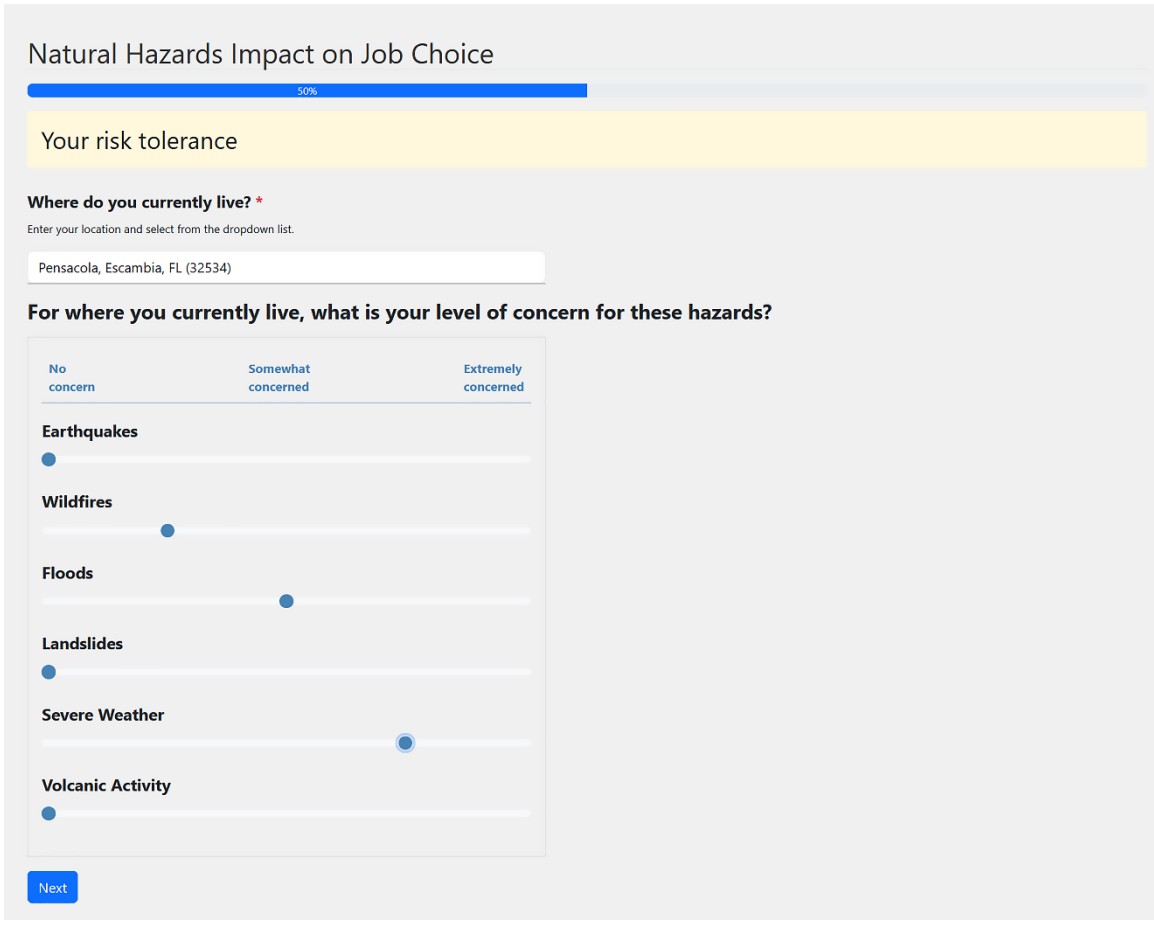


**Figure A6: Reflection on the concern for hazards in the participant's current location of residence (Gonzales et al., 2025).**



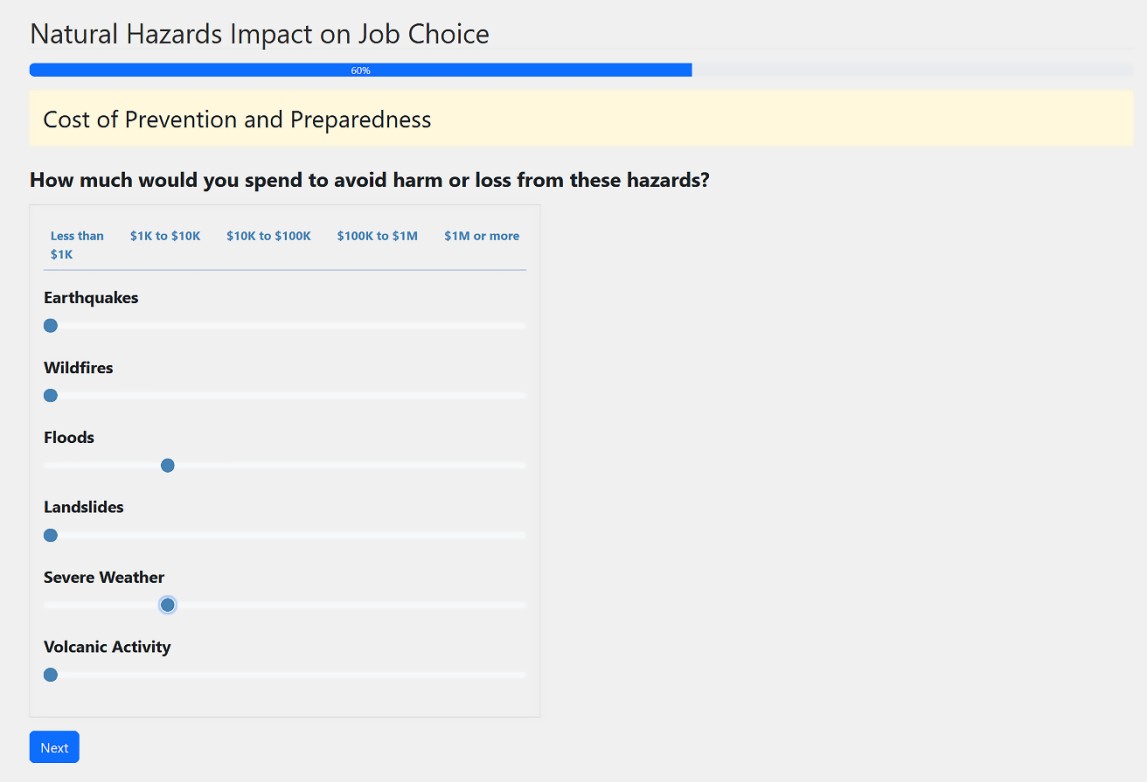

**Figure A7: Reflection on how much the participant would be willing to pay for hazard prevention and preparedness (Gonzales et al., 2025).**




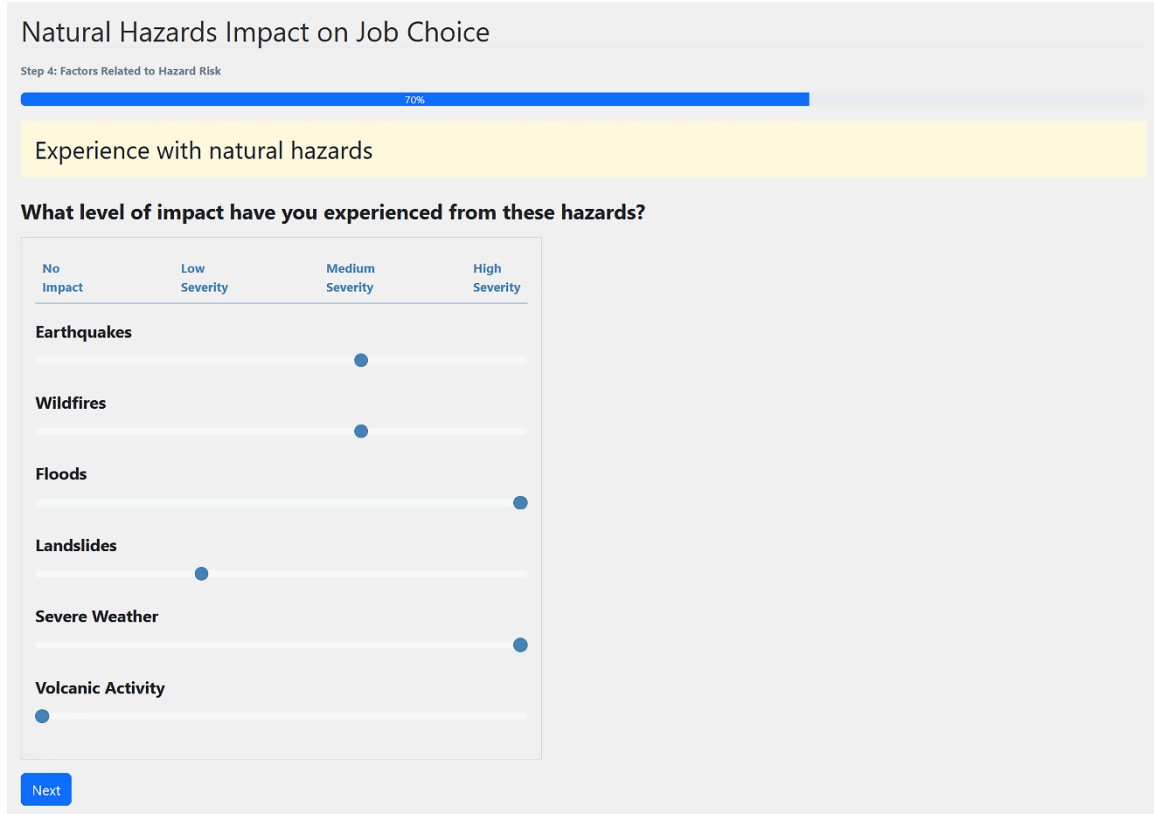

**Figure A8: Reflection on the experience with hazard impacts (Gonzales et al., 2025).**




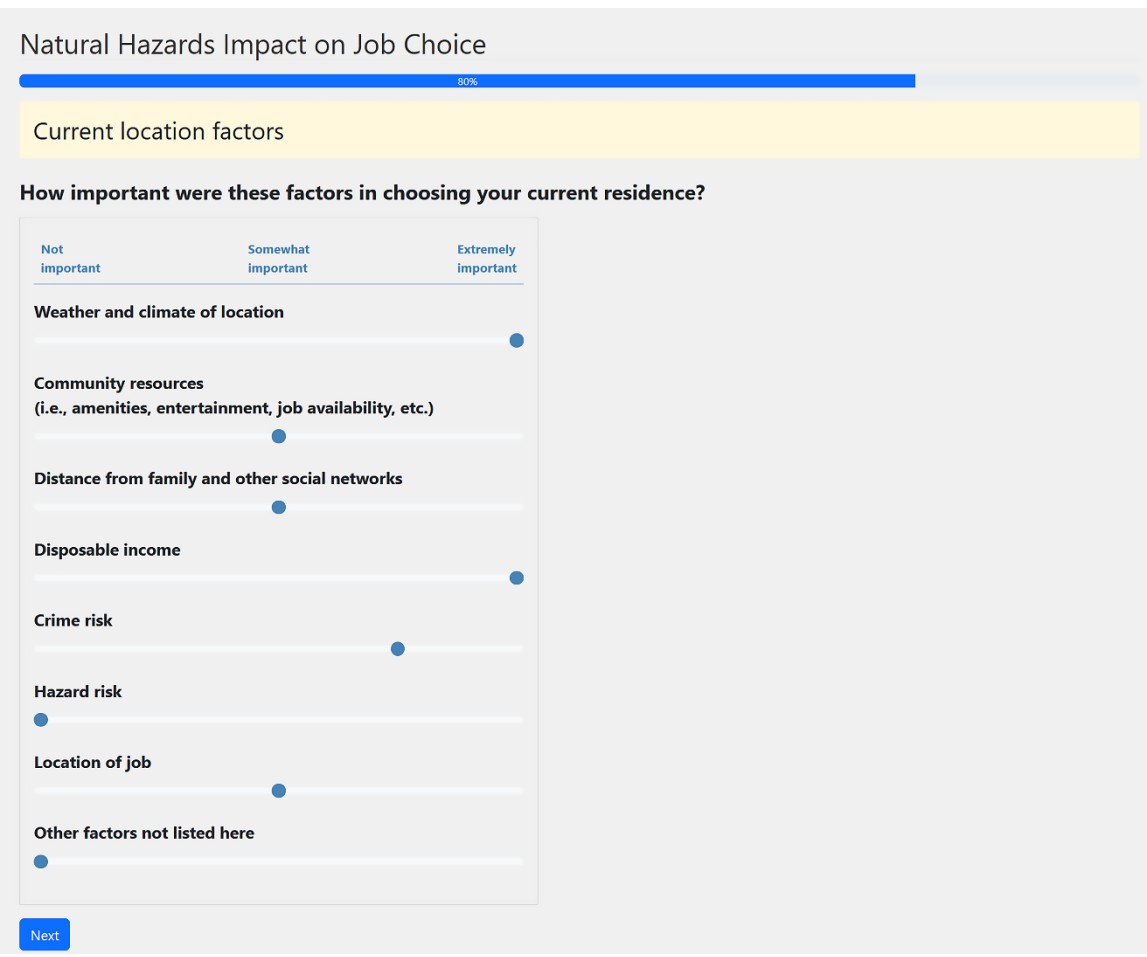


**Figure A9: Reflection on the factors important in the choice of the participant's current location (Gonzales et al., 2025).**

4000



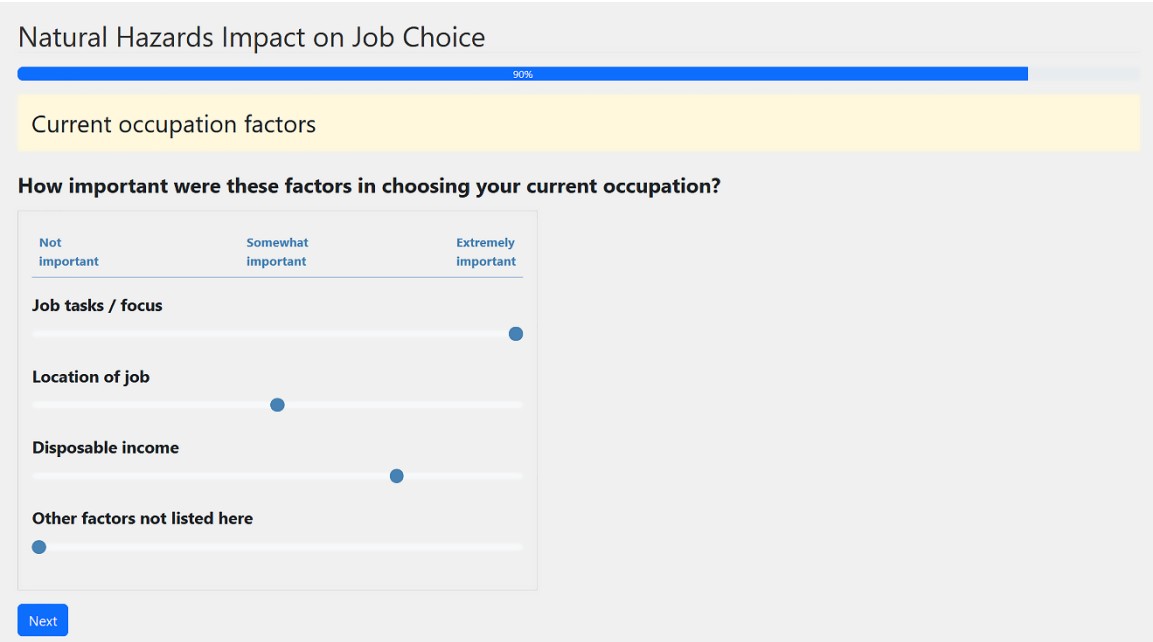

**Figure A10: Reflection on the factors important in the choice of the participant's current occupation (Gonzales et al., 2025).**


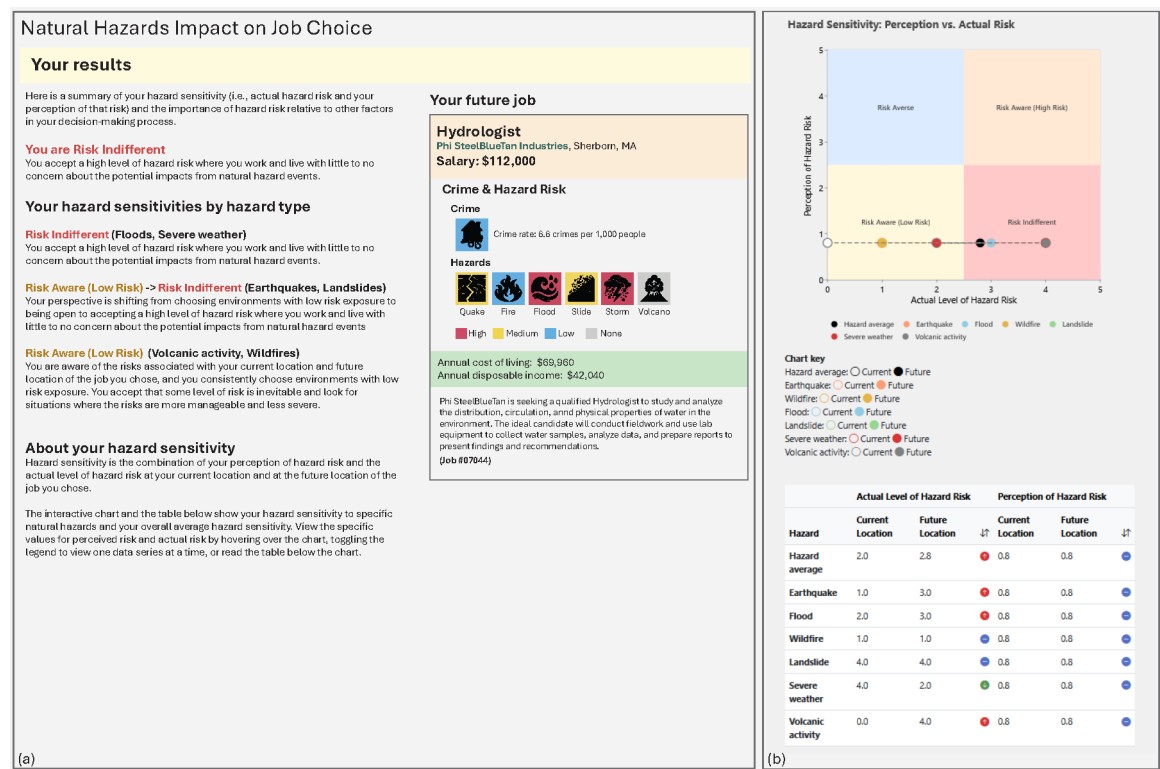





**Figure A11: Simulation results: Hazard sensitivity (Gonzales et al., 2025).**

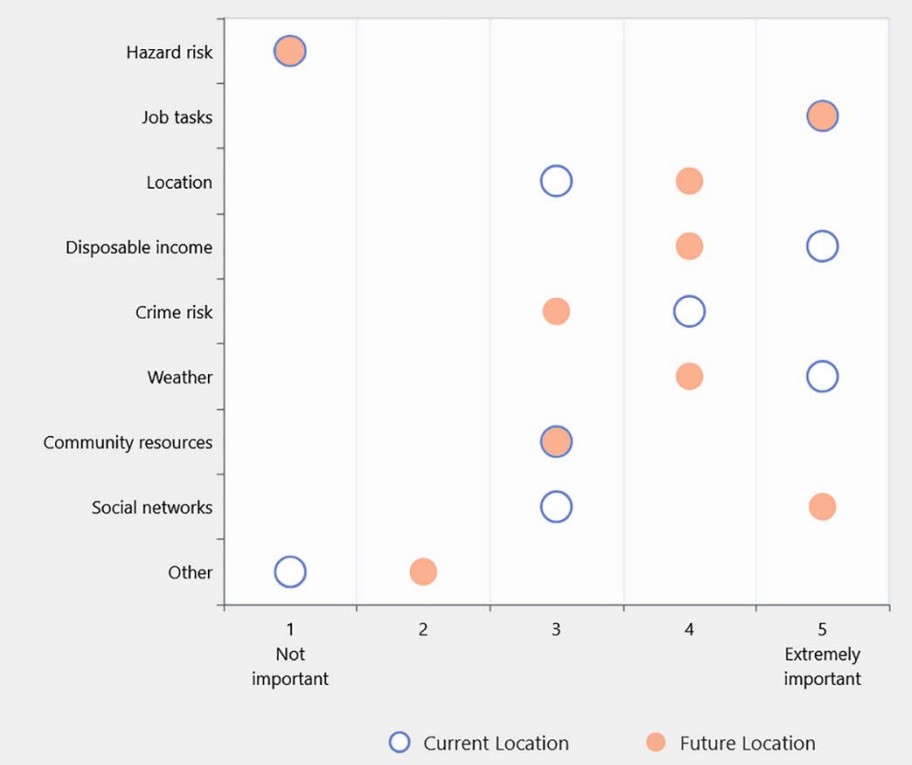

**Figure A12: Simulation results: Factors important to decision making (Gonzales et al., 2025).**



## Appendix B: Database Sources and Metadata

### US Bureau of Labor Statistics, Occupational Employment Statistics, May 2023

We used the state data as a base salary for a given occupation for each state, and the occupational profile descriptions as a base to generate degree field specific job descriptions and titles. Where salary information was marked at an upper limit, we used the maximum salary value presented in the table, which was $239,200 USD. Based on AGI's Status of the Geoscience Workforce report schema (https://profession.americangeosciences.org/reports/status-geoscience-workforce) for defining geoscience-related occupations, we flagged all geoscience related occupations in the dataset.

### 620 US Federal Emergency Management Agency, National Risk Index

We used the National Risk Index (NRI) data to map natural hazard risk index values to the county level for each state. We included the overall hazard risk value for the county and then mapped the following six hazard categories to specific hazards from the NRI dataset. For our individual hazard category values, we selected the highest hazard risk rating for all specific NRI hazards in that category.


| Category | Specific Hazards from NRI dataset |
|---|---|
| **Earthquake** | earthquake |
| **Fire** | wildfire |
| **Slide** | avalanche, landslide |
| **Flood** | coastal flood, riverine flood, tsunami |
| **Severe weather** | cold wave, drought, hail, heat wave, hurricane, ice storm, lightning, strong wind, tornado, winter weather |
| **Volcanic activity** | volcano |

**Table B1: Categorization of NRI hazard types for simulation.**

### Economic Policy Institute Family Budget Calculator, January 2024.

We used this data to calculate cost of living for a single person with no children at the county level. Data included in its cost-
of-living calculation are housing costs, food costs, childcare (not applicable to the single person no children category we used), transportation costs, health care costs (i.e. insurance premiums and out of pocket costs), other necessities (i.e., apparel, personal care, household supplies, etc.), and income and payroll taxes.

 Data are in 2023 dollars. https://www.epi.org/resources/budget/



**FBI Crime Data Explorer, NBIRS Tables: State tables, offenses by agency for 2022.**

Because we needed population data to calculate the crime rate (as crime / 1000 persons), we could only use the city level data. Note that only cities that provide data to the FBI are included in this dataset.

**O\*NET® 28.3 Database, U.S. Department of Labor, Employment and Training Administration**

We used with permission under the CC BY 4.0 license the job zone data from the O\*NET® 28.3 Database. The job zone data maps the standard occupational codes used by the US Bureau of Labor Statistics to educational achievement levels.

**US Housing & Urban Development-US Postal Service ZIP Code crosswalk**

This data was used to create a geocoded postal zip code, city, county lookup function for the simulation to aid users in
specifying their location of current residence.

**Tables and metadata**

**crime_hazards_cost_of_living_combined**

- US state, county, city
- US Census Bureau region and division
- Crime rate and rank
- Cost of living annual cost
- Hazard risk level, overall and per hazard

**jobZonesSalariesByState**

- Occupation code (detailed and broad)
- US state, US Census Bureau region and division
- Salary percentiles (10, 25, 50, 75, 90) for occupation in state
- Job zone – from ONET dataset. The minimum required level of educational attainment.


**jobDescriptions**

- Occupation code (detailed and broad)
- Degree field code and name
  - o Art and humanities
- o Biological and agricultural sciences
  - o Chemistry
  - o Computer and information sciences





- o Earth, atmospheric, and ocean sciences
- o Education
- o Engineering
- o Health
- o Management and administration
- o Mathematics and statistics
- o Physics and astronomy
- o Sales and marketing
- o Social sciences
- o Social service
- o Technology and technical fields
- o Non-Science & Engineering fields
- o Science & Engineering-related fields
- o Generic – to match on participants who did not specify a degree field.

- • Organization name (randomly assigned combination of a letter from the Greek alphabet and a Crayola crayon color name).
- • Job title – AI generated job title based on the degree field and base occupational title.
- • Job description – AI generated job description based on the degree field and base occupational description.

**nri_counties**
- • All columns from the FEMA NRI dataset for all counties.
- • We use the values from the State-County FIPS Code and the Hazard Type Risk Index Rating value for each specific
hazard.

**zipcode_city_county_crosswalk**
- • City name
- • Postal zip code
- • County name
- • US state abbreviation
- • State and County Federal Information Processing Standard (FIPS) code





**Acknowledgements**

We would like to thank all the people that took time out of their day to participate in the different aspects of the study. Furthermore, David Crookall, Giuseppe Di Capua, and Rachel Wellman are thanked for helpful comments.

**Code availability**

The career-choice risk simulation code can be accessed from https://github.com/AmericanGeosciencesInstitute/GRANDE-simulation.

**Data availability**

All the primary survey and simulation data that is not already provided in the paper and figures has been kept anonymous due to the ethics review recommendation (application no. AGIGRANDE-202205). The in-depth analysis of funding, curriculum, and scholarly research as well as direct survey results can be accessed on the project website at https://grande.americangeosciences.org.

**Author contribution**

LG and CK conceptualized, designed, and implemented the study components. RB helped with the analysis of the study results and provided input in the implementation and analysis of the hazard choice game. LG prepared the manuscript with contributions from all co-authors.

**Competing interests**

The authors declare that they have no conflict of interest.

**Ethical statement**

As discussed in the Methods section, we submitted an ethics application to the Advarra Institutional Review Board (AGIGRANDE-202205). The project study was exempted from IRB oversight.

**Financial support**

Funding for this project is provided by the National Science Foundation (Award #2223004).



## Disclaimer

The results and interpretations are the views of the American Geosciences Institute and not those of the National Science Foundation.

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
