# Peer review of "The Gap Between Attitudes and Action Within the US Geoscience Community's Response to Natural Hazards"

_EGUsphere, 2025_

## Author Response (AR1)

**Author's response to comments from editor and reviewers**

**Editor Comments**

1) The region-specific nature of your manuscript should be reflected in the title.

**We have updated the title to read "The Gap Between Attitudes and Action Within the US Geoscience Community's Response to Natural Hazards".**

2) Since you have chosen not to engage with Reviewer 2's comment regarding the political ecology of expertise—which is, of course, your prerogative—you should clearly acknowledge this as a limitation of the study.

**We have added a sentence to the Limitations and Future Directions to note that this paper does not address the political and epistemic aspects of the knowledge action gap, and that additional research could provide more insights into areas where interventions may help to address the gap.**

**RC1: ['Comment on egusphere-2025-3430'](), Hazel Napier, 07 Nov 2025**

it is not clear in the abstract where in the world this study is focused. Later it is made clear it is US focused. This would be useful to uinclude further up the manuscript.

**We updated the abstract to specify that the study is US-focused.**

Suggest that it is made clear in the abstract that the job-choice simluation was used and forms the bulk of the paper. This would provide useful clarity when reading early sections of the paper (introductino etc.).

**We updated the abstract to clarify that the job-choice simulation forms the bulk of the paper.**

It would be useful to understand the size of the surveys conducted between 2023-2025. Who was surveyed (how many academics/researchers etc) and how many institutions. Perhaps also show results of the survey at p7 in a table.

**We provided as much information as possible on sub-cohort sizes for each of the surveys in start of section 2.3. Since we provided the detail in the text, we decided against repeating the information in an additional table.**

The reasoning for using the online job-choice simluation is sound however it is unclear where participants were drawn from. How wide was the cohort and how representative of the academic/non-academic population as a whole?

**We added text to p14 to briefly describe our recruitment strategies for the simulation and to clarify that we were not aiming for a representative sample of the US population. In the Results and Discussion section we further clarify our sampling in that the focus of the simulation was on discipline-wide cohorts rather than occupational cohorts, such as academic vs. non-academic populations.**

P19-20, lines 420 and 421 - sentence unclear - 'Thus in reflection, the importance of income may have become less important than income'

**Thank you for catching that. The sentence should have read 'Thus in reflection, the importance of income may have become less important than other factors, such as location, community amenities, and favourable weather'. We have updated the text accordingly.**

Suggest some of the results from the job-choice simulations are presented in tables. The text is dense and hard to read at times. Some form of summary of the results would be useful (there is some summary information in the form of charts (figures 6 and 7), but tables may help the reader understand the key messages).

**We reviewed the text in the Results and Discussion section and noted that most paragraphs refer to either tables or charts that provide summaries of the data being discussed. We updated the first paragraph on p22 (L465) to refer the reader to Figure 7 which can be used alongside the text to better follow the discussion.**

**RC2**: 'Comment on egusphere-2025-3430', Anonymous Referee #2, 11 Nov 2025

Referee's verbatim comments are in plain font, followed by authors' replies in **bold**.

**Thank you for the feedback and suggestions to expand upon this paper's research. It is outside our scope of this research project to delve into the political and epistemological aspects of the knowledge-action gap. However, we recognize that**

**there are many opportunities for additional research and inquiry for which this paper sets the groundwork.**

**In response to some of the more specific comments:**

1. **The comment about the "patchwork of local initiatives" referred to text in the Abstract where we did not wish to include references to specific examples. We provide in the Introduction section citations for further reading regarding examples of localized initiatives for adaptation.**
2. **Institutional vulnerability. Thank you for the comments on the situation of geoscience departments in the United Kingdom. Our study is US-based and so we focus our paper and discussion on the situation of geoscientists engagement with hazards in the US. Further research into the evaluation of this knowledge-action gap would be useful, especially in other countries, such as the United Kingdom as is noted in the comment.**
3. **Integration of knowledge into personal decision-making. The paper specifically refers to integration of knowledge as it pertains to the results of the job-choice simulation in which geoscientists and non-geoscientists showed similar patterns of interaction with the simulation.**

The paper provides an empirical analysis of how geoscientists engage with, or fail to engage with, natural hazard risk both professionally and personally. The simulation that tests how hazard perception influences job choice is innovative and provides valuable methodological insight. The analysis exposes an enduring knowledge–action gap across the discipline, yet the framing of this gap remains primarily behavioural and technocratic. The discussion can be deepened by engaging directly with the political and epistemic dimensions of inaction, as well as with the institutional structures that make the gap systemic rather than individual.

The paper describes adaptation efforts as "a patchwork of local initiatives." This observation is valuable, but it would be more compelling if illustrated with a few clear examples. Cases of localised flood control or wildfire response could demonstrate how fragmented initiatives obscure the institutional or political nature of the problem. Work by Andrea Nightingale and Ritodhi Chakraborty (below) suggests that such patchworks often arise not from a lack of knowledge or funding but from excessive faith in technical and managerial fixes that depoliticise adaptation and obscure unequal power relations. Relating the geoscience community's response to this broader critique would shift the

analysis away from behavioural explanations of the knowledge–action gap and towards its structural and political causes.

The discussion of institutional vulnerability in the background section could also be extended. Geoscience departments have been politically and financially weakened in many countries, including the United Kingdom. They are particularly exposed at a time when climate change demands their expertise. Attention to the political economy of underfunding would help explain the discipline's uneven capacity to act. The decline of state support for science, the rise of market-oriented funding and the prioritisation of short-term, applied research all shape how geoscientists engage with hazards. Linking these trends to broader debates about neoliberal governance of science would strengthen the argument and situate the findings in a global academic context.

The section on integration could be developed further. The authors refer to integrating expert hazard knowledge into decision-making, but it is unclear what kinds of inclusion are involved and whether this process creates friction. Integration is rarely neutral: it often reinforces disciplinary hierarchies and epistemic inequalities. Scholarship on plural knowledge systems has shown that integration can sometimes undermine rather than promote justice, particularly when other perspectives are incorporated instrumentally rather than collaboratively. Clarifying how geoscientists integrate knowledge, whether through genuine co-production or through technical synthesis, would enhance the discussion of justice and inclusion within the discipline.

Overall, the paper makes a valuable contribution by documenting the systemic character of disengagement in geoscience. The findings reveal the limits of awareness and the persistence of cognitive dissonance within the community. However, the argument could reach further by engaging with the political ecology of expertise. The knowledge–action gap should be seen not only as an individual failure to act, but also as a structural outcome of how knowledge, institutions, and funding regimes are organised. Drawing on critiques of technocratic adaptation and on work about plural and situated knowledges (reading suggested) would show that the gap is not simply cognitive but political and epistemic. This would position geoscience not only as a discipline under strain but also as a site with the potential to transform how hazards and risks are understood and addressed.

Suggested Readings

Andrea J. Nightingale (2017). Power and Politics in Climate Change Adaptation Efforts: Struggles over Authority and Recognition in the Context of Political Instability. Geoforum, 84, 11–20.

Ritodhi Chakraborty, Mabel D. Gergan, Pasang Y. Sherpa & Costanza Rampini (2021). A Plural Climate Studies Framework for the Himalayas. Current Opinion in Environmental Sustainability, 49, 22–29.

Siri H. Eriksen, Andrea J. Nightingale & Hallie Eakin (2015). Reframing Adaptation: The Political Nature of Climate Change Adaptation. Global Environmental Change, 35, 523–533.

Philip Mirowski (2011) Science-Mart: Privatizing American Science, Harvard University Press.

Daniel Sarewitz (2016) Saving Science, The New Atlantis, 49, 4–40.

Sheila Jasanoff (2004) States of Knowledge: The Co-production of Science and the Social Order, Routledge.